# Bridging Protein Sequences and Microscopy Images with Unified Diffusion Models

**Dihan Zheng** [1]   **Bo Huang** [1 2 3]

## Abstract

Fluorescence microscopy is ubiquitously used in cell biology research to characterize the cellular role of a protein. To help elucidate the relationship between the amino acid sequence of a protein and its cellular function, we introduce CELL-Diff, a unified diffusion model facilitating bidirectional transformations between protein sequences and their corresponding microscopy images. Utilizing reference cell morphology images and a protein sequence, CELL-Diff efficiently generates corresponding protein images. Conversely, given a protein image, the model outputs protein sequences. CELL-Diff integrates continuous and diffusion models within a unified framework and is implemented using a transformer-based network. We train CELL-Diff on the Human Protein Atlas (HPA) dataset and fine-tune it on the Open-Cell dataset. Experimental results demonstrate that CELL-Diff outperforms existing methods in generating high-fidelity protein images, making it a practical tool for investigating subcellular protein localization and interactions.

## 1. Introduction

Protein sequences inherently encode their functions, and predicting these functions solely from sequence information has become a critical area of research. With the development of artificial intelligence, learning-based methods are increasingly employed to predict a wide range of protein properties, including structural conformation (Jumper et al., 2021; Baek et al., 2021), interaction partners (Evans et al., 2021), and subcellular localization (Almagro Armenteros et al., 2017; Khwaja et al., 2024b). Concurrently, the rapid

development of generative models has enabled researchers to design functional proteins (Madani et al., 2023; Dauparas et al., 2022) and drug-like molecules (Isigkeit et al., 2024). These computational methods allow for large-scale virtual screening, significantly reducing the costs and resources associated with experimental validation. The advent of those technologies presents significant opportunities for biomedical research, potentially accelerating advancements in therapeutic target identification, drug discovery, and the investigation of biochemical pathways (Palma et al., 2012).

In this work, we focus on the relationship between protein sequences and their cellular functions as characterized by microscopy images. Specifically, we focus on fluorescence microscopy, which is ubiquitously used in nearly all cell biology research. Fluorescence microscopy images provide extremely rich information for proteins of interest in the cellular context, such as their expression level, subcellular distribution, and molecular interactions as can be measured by spatial colocalization. Such information characterizes protein functional activities as well as the physiological and pathological state of cells. Disease-causing genetic mutations can alter the amino acid sequence of proteins, resulting in changes in image phenotypes by modifying gene expression patterns, reshaping molecular interaction profiles, or globally perturbing cellular states. As a first step towards building a model that connects the sequence of proteins and their cellular images, recently, Khwaja et al. (2024b) proposed CELL-E, a text-to-image transformer that predicts fluorescence protein images from sequence input and cell morphology condition. Furthermore, CELL-E2 (Khwaja et al., 2024a) was developed to enhance the image generation speed of CELL-E by utilizing the idea of masked token prediction (Chang et al., 2022). Additionally, CELL-E2 facilitates the bidirectional transformation between sequences and images. However, their image model only allowed outputs of highly blurred images lacking fine details to discern any of the subcellular structures other than the most prominent one (i.e., the nucleus), restricting their applicability only to the study of a very limited set of sequence features (i.e., the nuclear localization signal).

To expand the application of sequence-to-cell-image generative models, we introduce CELL-Diff, a unified diffusion model that enables bidirectional transformation between

---

[1]Department of Pharmaceutical Chemistry, UCSF, San Francisco, CA 94143 [2]Department of Biochemistry and Biophysics, UCSF, San Francisco, CA 94143 [3]Chan Zuckerberg Biohub San Francisco, San Francisco, CA 94158. Correspondence to: Bo Huang <bo.huang@ucsf.edu>.

*Proceedings of the 42^{st} International Conference on Machine Learning*, Vancouver, Canada. PMLR 267, 2025. Copyright 2025 by the author(s).

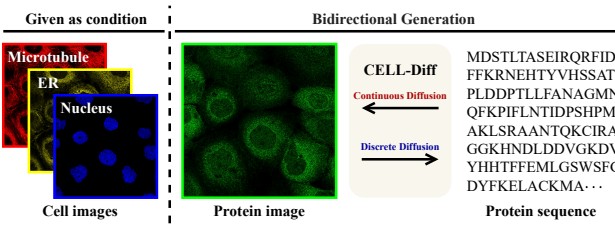

*Figure 1.* Given cell morphology images as conditional input, CELL-Diff facilitates bidirectional generation between protein sequences and images.

protein sequences and their corresponding microscopy images. Specifically, by utilizing cell morphology images including the nucleus and cytoplasmic markers such as endoplasmic reticulum (ER) and microtubules as conditional input, CELL-Diff can generate detailed protein images from given protein sequences. Conversely, it can also output protein sequences when provided with microscopy images, as shown in Figure 1. To enable this bidirectional transformation, CELL-Diff employs the continuous diffusion model for generating microscopy images and the discrete diffusion model for redesigning protein sequences, which can be integrated within a unified framework. Specifically, we design a transformer-based U-Net model (Ronneberger et al., 2015; Peebles & Xie, 2023) enhanced with the cross-attention mechanism from Stable Diffusion (Rombach et al., 2022). This model processes sequences and images as inputs, simultaneously modeling the conditional distributions for image-to-sequence and sequence-to-image transformations. The final objective function comprises the noise prediction loss for the continuous diffusion model and the masked value prediction loss for the discrete diffusion model. We evaluate CELL-Diff on HPA dataset (Digre & Lindskog, 2021), which provides cellular microscopy images of human proteins based on fixed immunofluorescence staining. Subsequently, we fine-tune the model on the Open-Cell dataset (Cho et al., 2022), which offers live microscopy images of different human cell lines, each tagged with a single protein via CRISPR/Cas9 gene editing.

- We present CELL-Diff, a diffusion-based generative model that enables conditional bidirectional generation of protein sequences and their corresponding microscopy images. By integrating the continuous diffusion and discrete diffusion models, CELL-Diff can be trained within a unified framework. We propose a transformer-based U-Net model for implementing CELL-Diff, which effectively integrates information from images and sequences.

- We train CELL-Diff on the HPA dataset using different conditional cell images and fine-tune it on the Open-

Cell dataset. Experimental results show that our model generates more detailed and sharper protein images than previous methods.

## 2. Related works

Multi-modal generative modeling can be formalized as learning the conditional or joint distribution between modalities. Representative applications include text-to-image generation (Ramesh et al., 2021; Ding et al., 2021; Nichol et al., 2022), image-to-text generation (image captioning) (Mokady et al., 2021; Chen et al., 2023), text-to-video generation (Ho et al., 2022), and text-to-speech (Chen et al., 2021; Popov et al., 2021). Most of these approaches rely on diffusion models or auto-regressive models for the generation and typically focus on unidirectional transformation. However, our goal is to achieve bidirectional generation, which requires learning joint distributions. To achieve this, Hu et al. (2023) proposed a discrete diffusion-based model for learning the joint distribution between images and text, though its scalability remains unexplored. Bao et al. (2023) introduced Unidiffuser, a unified diffusion model capable of unconditional, conditional, and joint generation. The key observation of Unidiffuser is that the learning objective of the diffusion score function can be unified in a general framework with multiple diffusion time steps. Furthermore, Zhou et al. (2024) developed Transfusion, which integrates auto-regressive and diffusion models for both single and cross-modality generation. Transfusion combines the auto-regressive loss with diffusion, training a single transformer model using an extended causal mask. These methods generally depend on large pre-trained encoders for images and text, such as CLIP (Radford et al., 2021) and VQGAN (Esser et al., 2021). However, for microscopy images, the variability in equipment and experimental conditions limits the availability of such robust image encoders, making the direct application of these models challenging. Indeed, the two previous protein-sequence-to-microscopy generators, CELL-E (Khwaja et al., 2024b) and CELL-E2 (Khwaja et al., 2024a), which both used VQGAN, only produce coarse-grain images that have too much blur to distinguish fine-scale subcellular structures such as the cytoskeleton. As for CELL-Diff, we combine continuous and discrete diffusion to enable bidirectional transformation between protein images and sequences, which achieves high-fidelity image generation while preserving the ability to resolve detailed subcellular structures.

## 3. Technical background

Before delving into our unified diffusion model, we briefly introduce the background of diffusion models applicable to continuous and discrete state spaces. Specifically, we employ the continuous and discrete diffusion models for

generating microscopy images and protein sequences, respectively.

## 3.1. Diffusion model for continuous state spaces

Let $\mathbf{I}_0$ be a continuous random variable in $\mathbb{R}^d$, where $d$ denotes the dimension, and let $\mathbf{I}_{1:T} = \{\mathbf{I}_t\}_{t=1}^T$ be a sequence of latent variables, with $t$ as the index for diffusion steps. The diffusion model involves two processes: the forward process and the reverse process. In the forward process, the diffusion model progressively injects noise into the initial data $\mathbf{I}_0$, transforming it into a Gaussian random variable $\mathbf{I}_T$. In the reverse process, the model learns to invert the diffusion process through a denoising model and generate new data by gradually eliminating the noise.

**Forward process.** The forward process involves injecting noise into the initial data. Given a variance schedule $\{\beta_t\}_{t=1}^T$, the forward process is defined as:

$$q(\mathbf{I}_t|\mathbf{I}_{t-1}) = \mathcal{N}(\sqrt{1-\beta_t}\mathbf{I}_{t-1}, \beta_t \boldsymbol{I}_d), \quad t = 1, \ldots, T. \tag{1}$$

Let $\alpha_t = 1 - \beta_t$ and $\bar{\alpha}_t = \prod_{i=1}^t \alpha_i$. Then, for any arbitrary time step $t$, it holds that $q(\mathbf{I}_t|\mathbf{I}_0) = \mathcal{N}(\sqrt{\bar{\alpha}_t}\mathbf{I}_0, (1-\bar{\alpha}_t)\boldsymbol{I}_d)$. Consequently, for a sufficiently large $T$, this process will transform $\mathbf{I}_0$ into an isotropic Gaussian variable.

**Reverse process.** The goal of the reverse process is to generate new samples from $p(\mathbf{I}_0)$ starting from a Gaussian random variable $\mathbf{I}_T \sim \mathcal{N}(0, \boldsymbol{I}_d)$. The reverse process is defined by a Markov Chain with trainable transitions:

$$p_\theta(\mathbf{I}_{t-1}|\mathbf{I}_t) = \mathcal{N}(\mu_\theta(\mathbf{I}_t, t), \sigma_t^2 \boldsymbol{I}_d), \quad t = 1, \ldots, T. \tag{2}$$

Here, $\mu_\theta$ represents parameterized neural networks designed to estimate the means from the current state, and $\sigma_t^2$ denotes the variance.

**Training objective.** The training objective function can be derived using variational inference. Instead of optimizing the intractable log-likelihood function $\log p(\mathbf{I}_0)$, the diffusion model maximize its ELBO:

$$\mathbb{E}_q \Bigg[ \log p_\theta(\mathbf{I}_0|\mathbf{I}_1) - D_{\mathrm{KL}}(q(\mathbf{I}_T|\mathbf{I}_0) \| p_\theta(\mathbf{I}_T))$$
$$- \sum_{t=2}^T D_{\mathrm{KL}}\left( q(\mathbf{I}_{t-1}|\mathbf{I}_t, \mathbf{I}_0) \| p_\theta(\mathbf{I}_{t-1}|\mathbf{I}_t) \right) \Bigg], \tag{3}$$

where $q(\mathbf{I}_{t-1}|\mathbf{I}_t, \mathbf{I}_0)$ has an formulation as $\mathcal{N}(\frac{\sqrt{\bar{\alpha}_{t-1}}\beta_t}{1-\bar{\alpha}_t}\mathbf{I}^0 + \frac{\sqrt{\alpha_t}(1-\bar{\alpha}_{t-1})}{1-\bar{\alpha}_t}\mathbf{I}^t, \frac{(1-\bar{\alpha}_{t-1})\beta_t}{1-\bar{\alpha}_t}\boldsymbol{I}_d)$. Combining with (2), the KL divergence of two Gaussian distributions has a closed-form formulation.

To further simplify the computation, Ho et al. (2020) proposed a training objective based on a variant of the ELBO in (3) as

$$L_{\mathrm{DDPM}} = \mathbb{E}_{\mathbf{I}_0, t, \epsilon} \|\epsilon_\theta(\sqrt{\bar{\alpha}_t}\mathbf{I}_0 + \sqrt{1-\bar{\alpha}_t}\epsilon, t) - \epsilon\|_2^2, \tag{4}$$

where $\epsilon \sim \mathcal{N}(\mathbf{0}, \boldsymbol{I}_d)$ and $\epsilon_\theta$ is a noise prediction network. In this formulation, $\mu_\theta$ can be parameterized by $\epsilon_\theta$, enabling the generation of new samples through the reverse process.

## 3.2. Diffusion model for discrete state spaces

Several distinct diffusion models are designed for discrete data (Austin et al., 2021; Hoogeboom et al., 2021). This section focuses on the order-agnostic Autoregressive Diffusion Models (OA-ARDM) (Hoogeboom et al., 2022).

Let $\mathbf{S} = (\mathbf{S}_1, \ldots, \mathbf{S}_D)$ be a multivariate random variable, where $D$ denotes the maximum sequence length, $\forall t \in \{1, \ldots, D\}$, $\mathbf{S}_t \in \{1, \ldots, K\}$ with $K$ categories. Denote $S_D$ as the set of all permutations of $\{1, \ldots, D\}$, and assume $\sigma$ represents a random ordering in $S_D$. Applying Jensen's inequality, we obtain:

$$\log p(\mathbf{S}) = \log \mathbb{E}_{\sigma \sim \mathcal{U}(S_D)} p(\mathbf{S}|\sigma) \geq \mathbb{E}_{\sigma \sim \mathcal{U}(S_D)} \log p(\mathbf{S}|\sigma), \tag{5}$$

where $\mathcal{U}(S_D)$ denotes the uniform distribution over $S_D$. Following order $\sigma$, $\log p(\mathbf{S}|\sigma)$ can be factorized as $\sum_{t=1}^D \log p(\mathbf{S}_{\sigma(t)}|\mathbf{S}_{\sigma(<t)})$, where $\mathbf{S}_{\sigma(<t)} = (\mathbf{S}_{\sigma(1)}, \ldots, \mathbf{S}_{\sigma(t-1)})$. Combining this with (5), we have:

$$\log p(\mathbf{S}) \geq \mathbb{E}_{\sigma \sim \mathcal{U}(S_D)} \sum_{t=1}^D \log p(\mathbf{S}_{\sigma(t)}|\mathbf{S}_{\sigma(<t)})$$
$$= \mathbb{E}_{\sigma \sim \mathcal{U}(S_D)} \sum_{t=1}^D \frac{\sum_{k \in \sigma(\geq t)} \log p(\mathbf{S}_k|\mathbf{S}_{\sigma(<t)})}{D - t + 1}. \tag{6}$$

Therefore, denote $\mathbf{f}_\theta$ as the neural network, $\mathcal{C}$ as the categorical distribution, the loss function for OA-ARDM is

$$L_{\mathrm{OA\text{-}ARDM}} = \mathbb{E}_{\mathbf{S}, t, \sigma} \frac{\sum_{k \in \sigma(\geq t)} -\log \mathcal{C}(\mathbf{S}_k|\mathbf{f}_\theta(\mathbf{S}_{\sigma(<t)}))}{D - t + 1}, \tag{7}$$

where $\sigma \sim \mathcal{U}(S_D)$ and $t \sim \mathcal{U}(1, \ldots, D)$.

The objective function of OA-ARDM corresponds to the "Masked Language Modeling" training objective proposed in BERT (Kenton & Toutanova, 2019) with a reweighting term. At each training step, we first sample a time step $t$ from $\mathcal{U}(1, \ldots, D)$, followed by a random ordering $\sigma$ from $\mathcal{U}(S_D)$. We then feed $\mathbf{S}_{\sigma(<t)}$ into the model, which predicts the remaining values $\mathbf{S}_{\sigma(\geq t)}$. In the generation step, we first sample a random ordering and then generate the values according to that order. These processes are facilitated through a masking operation, see Appendix A for the details.

# 4. Methodology

In this section, we introduce our unified diffusion model for generating microscopy images and protein sequences. Let $\mathbf{I}^{\mathrm{prot}}$ represent the protein image, $\mathbf{I}^{\mathrm{cell}}$ represent the cell morphology image, and $\mathbf{S}$ represent the protein sequence.

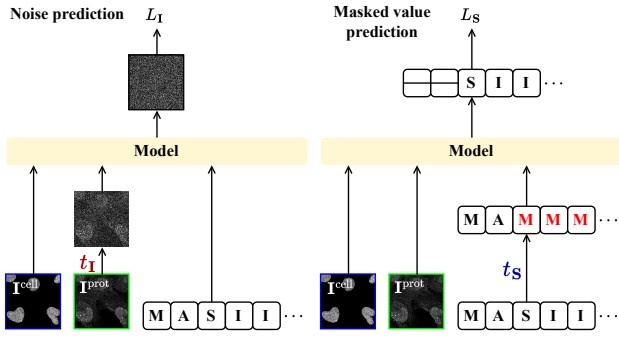

*Figure 2.* Training losses of CELL-Diff. During each training iteration, the protein image $\mathbf{I}^{\text{prot}}$ and sequence $\mathbf{S}$ are transformed using the forward processes of the continuous and discrete diffusion models, with randomly sampled time steps $t_{\mathbf{I}}$ and $t_{\mathbf{S}}$, respectively. The network model is tasked with predicting the noise in the protein image and the masked values from the protein sequence, corresponding to the noise prediction loss $L_{\mathbf{I}}$ and the masked value prediction loss $L_{\mathbf{S}}$.

The task of protein image prediction involves sampling from the conditional distribution $p(\mathbf{I}^{\text{prot}}|\mathbf{S}, \mathbf{I}^{\text{cell}})$, while the task of sequence generation involves sampling from $p(\mathbf{S}|\mathbf{I}^{\text{prot}}, \mathbf{I}^{\text{cell}})$. To achieve these goals within a unified diffusion model, we simultaneously model these two conditional distributions, which involves modeling a continuous variable $\mathbf{I}^{\text{prot}}$ and a discrete variable $\mathbf{S}$.

### 4.1. Proposed method

To simultaneously model $p(\mathbf{I}^{\text{prot}}|\mathbf{S}, \mathbf{I}^{\text{cell}})$ and $p(\mathbf{S}|\mathbf{I}^{\text{prot}}, \mathbf{I}^{\text{cell}})$, we aim to maximize the conditional log-likelihoods $\log p(\mathbf{I}^{\text{prot}}|\mathbf{S}, \mathbf{I}^{\text{cell}})$ and $\log p(\mathbf{S}|\mathbf{I}^{\text{prot}}, \mathbf{I}^{\text{cell}})$.

For $\log p(\mathbf{I}^{\text{prot}}|\mathbf{S}, \mathbf{I}^{\text{cell}})$, based on the continuous diffusion model assumption described in Section 3.1, we introduce latent variables $\mathbf{I}_{1:T} = \{\mathbf{I}_t\}_{t=1}^{T}$. Following the training objective of continuous diffusion (3), the conditional ELBO for $\log p(\mathbf{I}^{\text{prot}}|\mathbf{S}, \mathbf{I}^{\text{cell}})$ is expressed as:

$$\mathbb{E}_q\left[\log p_\theta(\mathbf{I}^{\text{prot}}|\mathbf{I}_1, \mathbf{S}, \mathbf{I}^{\text{cell}}) - D_{\text{KL}}(q(\mathbf{I}_T|\mathbf{I}_0)\|p_\theta(\mathbf{I}_T))\right.$$
$$\left. - \sum_{t_{\mathbf{I}}=2}^{T} D_{\text{KL}}\left(q(\mathbf{I}_{t_{\mathbf{I}}-1}|\mathbf{I}_{t_{\mathbf{I}}}, \mathbf{I}^{\text{prot}})\|p_\theta(\mathbf{I}_{t_{\mathbf{I}}-1}|\mathbf{I}_{t_{\mathbf{I}}}, \mathbf{S}, \mathbf{I}^{\text{cell}}))\right)\right],$$
$$(8)$$

For $\log p(\mathbf{S}|\mathbf{I}^{\text{prot}}, \mathbf{I}^{\text{cell}})$, we follow the discrete diffusion model approach described in Section 3.2, introducing a random ordering $\sigma$ for the decomposition of the sequence $\mathbf{S}$. Based on (6), the conditional ELBO for $\log p(\mathbf{S}|\mathbf{I}^{\text{prot}}, \mathbf{I}^{\text{cell}})$

is formulated as:

$$\mathbb{E}_\sigma \sum_{t_{\mathbf{S}}=1}^{D} \frac{1}{D-t_{\mathbf{S}}+1} \sum_{k\in\sigma(\geq t_{\mathbf{S}})} \log p(\mathbf{S}_k|\mathbf{S}_{\sigma(<t)}, \mathbf{I}^{\text{prot}}, \mathbf{I}^{\text{cell}}),$$
$$(9)$$

where $\sigma \sim \mathcal{U}(S_D)$.

Using the same parametrization technique as shown in (4) and (7), let $\mathbf{f}_\theta$ represents the neural network. The training objective for protein images is defined as:

$$L_{\mathbf{I}} = \mathbb{E}_{\mathbf{I}^{\text{prot}}, \mathbf{I}^{\text{cell}}, \mathbf{S}, t_{\mathbf{I}}, \epsilon}\|\mathbf{f}_\theta(\mathbf{S}, \mathbf{I}_{t_{\mathbf{I}}}, t_{\mathbf{I}}, \mathbf{I}^{\text{cell}})) - \epsilon\|_2^2. \quad (10)$$

where $\mathbf{I}_{t_{\mathbf{I}}} = \sqrt{\bar{\alpha}_{t_{\mathbf{I}}}}\mathbf{I}^{\text{prot}} + \sqrt{1-\bar{\alpha}_{t_{\mathbf{I}}}}\epsilon$. The training objective for protein sequences is defined as:

$$L_{\mathbf{S}} = \mathbb{E}_{\mathbf{I}^{\text{prot}}, \mathbf{I}^{\text{cell}}, \mathbf{S}, t_{\mathbf{S}}, \sigma} \frac{\sum_{k\in\sigma(\geq t_{\mathbf{S}})} - \log \mathcal{C}(\mathbf{S}_k|\mathbf{\Phi})}{D-t_{\mathbf{S}}+1}, \quad (11)$$

where $\mathbf{\Phi} = \mathbf{f}_\theta(\mathbf{S}_{\sigma(<t_{\mathbf{S}})}, \mathbf{I}^{\text{prot}}, 0, \mathbf{I}^{\text{cell}})$.

Finally, by combining (10) and (11), and introducing a balancing coefficient $\lambda$, the total loss for the proposed CELL-Diff model is given by:

$$L_{\text{CELL-Diff}} = L_{\mathbf{I}} + \lambda L_{\mathbf{S}}. \quad (12)$$

The training strategy is illustrated in Figure 2. During the training phase, the protein image loss $L_{\mathbf{I}}$ is computed by applying the forward diffusion process to the image, where Gaussian noise is incrementally added. The model then takes the corrupted image and the corresponding protein sequence as inputs to predict the noise introduced into the protein image. Similarly, the sequence loss $L_{\mathbf{S}}$ is computed by partially masking the sequence using a randomly generated mask. The model uses the masked sequence and microscopy images as inputs to predict the masked values within the sequence.

### 4.2. Model details

**Latent diffusion model.** To reduce computational costs, we employ the latent diffusion model (Rombach et al., 2022; Peebles & Xie, 2023) for modeling microscopy images. Specifically, we first train a Variational Autoencoder (VAE) (Kingma, 2013) to compress microscopy images into lower-dimensional spatial representations. After training, the VAE model is fixed, and the CELL-Diff model is trained using the latent representations of microscopy images produced by the VAE encoder. During image generation, a latent representation is sampled from the diffusion model and subsequently decoded into a full-resolution image using the learned VAE decoder.

**Inference.** After training, we can generate samples from two conditional distributions: $p(\mathbf{I}^{\text{prot}}|\mathbf{S}, \mathbf{I}^{\text{cell}})$ and

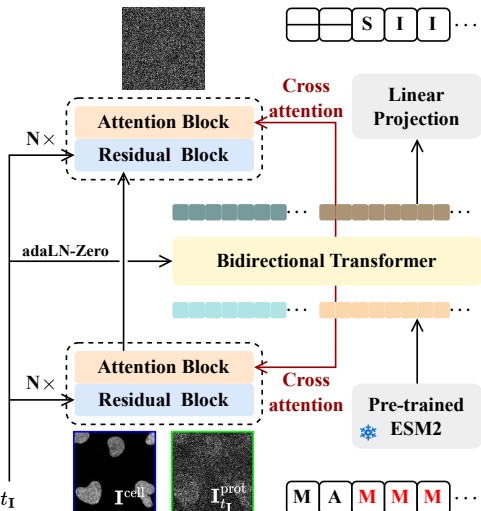

*Figure 3.* Network architecture of CELL-Diff. Microscopy images are embedded into a latent sequence through residual and attention blocks. The protein sequences are embedded using a pre-trained ESM2 model (Lin et al., 2022). These embeddings are concatenated and processed by a bidirectional transformer. The U-Net architecture (Ronneberger et al., 2015) is employed to output the noise in the protein image, while a linear projection is utilized to predict the masked values in the protein sequence. Cross-attention mechanisms are implemented to enhance information integration from images and sequences.

$p(\mathbf{S}|\mathbf{I}^{\text{prot}}, \mathbf{I}^{\text{cell}})$. Specifically, to generate the protein image $\mathbf{I}^{\text{prot}}$, we utilize the conventional reverse diffusion process as shown in (2), conditioning on the unmasked protein sequence $\mathbf{S}$ and the cell image $\mathbf{I}^{\text{cell}}$. The network model employed for generation is $\mathbf{f}_\theta(\mathbf{S}, \cdot, t_\mathbf{I}, \mathbf{I}^{\text{cell}})$, where $t_\mathbf{I} = 1, \ldots, T$. For the generation of the protein sequence $\mathbf{S}$, we utilize the reverse process of discrete diffusion OA-ARDM (Hoogeboom et al., 2022). We first sample a random ordering $\sigma$, and then generate sequence from $p(\mathbf{S}_{\sigma(t_\mathbf{S})}|\mathbf{S}_{\sigma(<t_\mathbf{S})}, \mathbf{I}^{\text{prot}}, \mathbf{I}^{\text{cell}})$, where $t_\mathbf{S} = 1, \ldots, D$. The network model in this scenario is $\mathbf{f}_\theta(\cdot, \mathbf{I}^{\text{prot}}, 0, \mathbf{I}^{\text{cell}})$. The sampling algorithm for OA-ARDM is shown in Algorithm 2.

**Network architecture.** As shown in (10) and (11), the network model $\mathbf{f}_\theta$ takes four inputs: the protein sequence $\mathbf{S}$, the protein image $\mathbf{I}^{\text{prot}}$, the cell image $\mathbf{I}^{\text{cell}}$, and the diffusion time step $t_\mathbf{I}$. To process the protein and cell images, we first concatenate them and then apply the commonly used U-Net architecture (Ronneberger et al., 2015). The concatenated images are fed into a series of downsampling blocks, transforming into image embeddings. The protein sequences are embedded using a pre-trained ESM2 model (Lin et al., 2022), with its parameters held fixed during training. Then, the image and protein embeddings are concatenated and processed using a bidirectional transformer. After passing through the transformer module, the concatenated feature

tensors are split into image and sequence feature tensors. The image feature tensors are then upsampled and combined with the downsampling features to output the noise from the protein image. The sequence feature tensor is processed using a linear projector to predict the masked values. The upsampling and downsampling blocks in the U-Net consist of residual and attention blocks. To enhance the integration of sequence information within the image processing component, we utilize cross-attention mechanisms with the attention blocks from Stable Diffusion (Rombach et al., 2022). Furthermore, we employ the adaptive layer norm zero (adaLN-Zero) conditioning method (Peebles & Xie, 2023) for incorporating the diffusion time step $t_\mathbf{I}$. The network architecture is illustrated in Figure 3.

## 5. Experiments

### 5.1. Datasets

**Human Protein Atlas.** The Human Protein Atlas (HPA) dataset (Digre & Lindskog, 2021) includes immunofluorescence images across various human cell lines with the proteins of interest stained by antibodies. It provides cellular images for 12,833 proteins and corresponding cell morphology images consisting of staining for the nucleus, ER, and microtubules. For each protein, the dataset includes multiple microscopy images from different cell lines. The corresponding protein sequences can be accessed from the UniProt dataset (UniProt Consortium, 2018). In total, we have collected 88,483 data points, each containing a protein sequence, a protein image, a nucleus image, an ER image, and a microtubule image.

**OpenCell.** The OpenCell (Cho et al., 2022) dataset provides a library of 1,311 CRISPR-edited HEK293T human cell lines, each with a target protein fluorescently tagged using the split-mNeonGreen2 system. For each target protein, OpenCell provides 4–5 confocal images along with a reference nucleus image. The cells were imaged live, offering a more accurate representation of protein distribution than the immunofluorescence images from HPA. Notably, 1,102 proteins are common between the HPA and OpenCell datasets. In total, we collected 6,301 data points, each containing a protein sequence, a protein image, and a nucleus image.

Given the size limitations of the HPA and OpenCell datasets, particularly in the diversity of protein sequences, we randomly selected 100 proteins from the shared subset between the two datasets as the test set, leaving the remainder for training. The test set for HPA and OpenCell contains 714 and 473 data points, respectively.

### 5.2. Implementation details

We begin by training a VAE with the same network architecture from Stable Diffusion on the training sets of the HPA

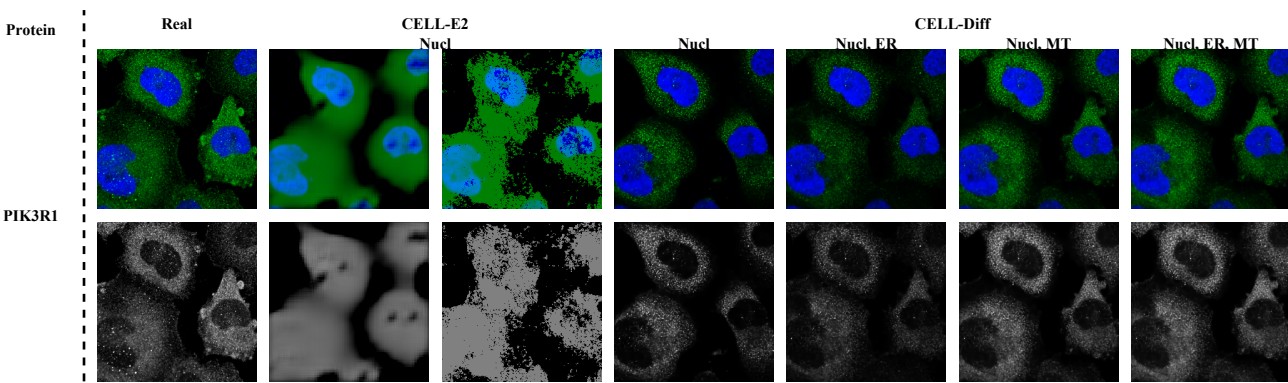

*Figure 4.* Visual results of protein image generation on HPA dataset.

*Table 1.* Comparison of protein image generation performance on HPA and OpenCell datasets. "Nucl" denotes the nucleus image, "ER" denotes the endoplasmic reticulum image, and "MT" denotes the microtubule image. "FID-T" indicates the FID computed using the thresholded protein image, and "FID-O" indicates the FID computed using the original protein image.

| Dataset | Method | Cell image | MSFR (nm) $\downarrow$ | IoU $\uparrow$ | FID-T $\downarrow$ | FID-O $\downarrow$ |
|---------|--------|-----------|------------------------|----------------|--------------------|--------------------|
| HPA | CELL-E2 | Nucl | 1824 | 0.462 | 77.6 | 166.4 |
|  | **CELL-Diff** | Nucl | **641** | **0.484** | **60.1** | **51.1** |
| HPA | **CELL-Diff** | Nucl, ER | 642 | 0.580 | 55.9 | 60.0 |
|  |  | Nucl, MT | 644 | 0.616 | 51.0 | 47.6 |
|  |  | Nucl, ER, MT | 644 | 0.635 | 50.4 | 45.6 |
| OpenCell | CELL-E2 | Nucl | 1165 | 0.466 | 77.0 | 245.8 |
|  | **CELL-Diff** | Nucl | **529** | **0.492** | **62.4** | **68.9** |

and OpenCell datasets for compressing microscopy images into a latent representation. For the HPA dataset, images are randomly cropped to a size of 1024 and then resized to 256, while images from the OpenCell dataset are directly cropped to 256 pixels. Data augmentation is applied using random flips and rotations. The latent representation has dimensions of $4 \times 64 \times 64$. The KL loss coefficient is set to $1 \times 10^{-5}$. The learning rate is initialized using a linear warm-up strategy, increasing from 0 to $3 \times 10^{-4}$ over the first 1,000 iterations, followed by a linear decay to zero. The batch size is set to 192. The VAE is trained for a total of 50,000 steps on the HPA dataset and fine-tuned for 20,000 steps on the OpenCell dataset.

Next, we fix the VAE model and train the latent diffusion model. CELL-Diff is pre-trained on the HPA dataset and fine-tune on the OpenCell dataset. Both pre-training and fine-tuning are conducted for 50,000 iterations using the Adam optimizer (Kingma & Ba, 2014). The learning rate is initialized using a linear warm-up strategy, increasing from 0 to $1 \times 10^{-4}$ over the first 1,000 iterations, followed by a linear decay to zero. The batch size is set to 64. The sequence embedding dimension is 1280, and the bidirectional transformer module consists of 8 layers with 8-head

attention. CELL-Diff is trained with 200 diffusion steps using the cosine noise schedules (Peebles & Xie, 2023), and use DDIM (Song et al., 2020) with 100 steps to accelerate the sampling speed. The weighting coefficient $\lambda$ in (12) is set to 1, and the maximum protein sequence length is 2,048. All models are trained using two Nvidia H200 GPUs.

**5.3. Protein image generation**

We evaluate the protein image generation performance of CELL-Diff. Given that the protein image prediction problem is relatively new, we compare CELL-Diff with the most closely related method, CELL-E2 (Khwaja et al., 2024a). The outputs for CELL-E2 are a heat map that indicates the localization distribution and a binary protein image, see Figure 4.

To provide a quantitative comparison, we introduce the Maximum Spatial Frequency Resolvability (MSFR) for microscopy images to measure its capability to discern fine structural details. Given a microscopy image $\mathbf{I}$, we define the Fourier Ring Power Spectral Density (FRPSD) as $\text{FRPSD}(r) = \sum_{r_i \in r} |\hat{\mathbf{I}}(r_i)|^2$, where $\hat{\mathbf{I}}$ denotes the Fourier transform of $\mathbf{I}$ and $r_i$ denotes the pixel element at radius $r$.

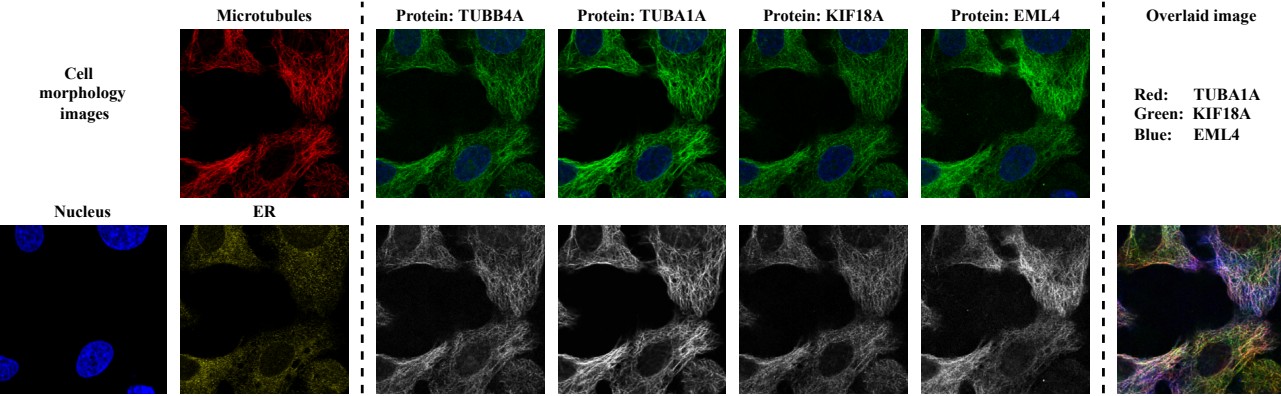

*Figure 5.* Virtual staining using HPA data. From identical cell images, CELL-Diff generates staining results for various proteins.

The MSFR is then defined as:

$$\text{MSFR} = \frac{1}{f}, \ f = \frac{i}{\text{Image Size} \times \text{Pixel Size}}, \quad (13)$$

where $i$ is defined as the first frequency index that satisfies:

$$\begin{cases} \text{FRPSD}(r) > 10^{-3}, & r < i \\ \text{FRPSD}(r) < 10^{-3}, & r = i \end{cases}. \quad (14)$$

We also employ the Intersection over Union (IoU) metric, which measures the similarity between two masks and is commonly used in image segmentation tasks. To calculate IoU, we apply median value thresholding to the original protein images to generate binary masks, while for CELL-E2, we use the predicted binary image. Additionally, we compute the Fréchet Inception Distance (FID) (Heusel et al., 2017) score to evaluate the similarity between the real and predicted images. FID is a learning-based metric that evaluates the quality of images generated by generative models. It measures the similarity between the generated and real images regarding their feature distributions. Lower FID scores indicate that the generated images are more similar to the real images. To compute FID, we concatenate the protein and nucleus images as input. In practice, we compute FID-T and FID-O, representing the FID score based on thresholding and original protein images, respectively. The results are shown in Table 1. The results show that CELL-Diff generated images exhibit better MSFR than CELL-E2. In particular, the MSFR for the original HPA and OpenCell data are 640 nm and 463 nm, respectively. The results from CELL-Diff are approaching the resolvability of the original training data, allowing us to discern finer details in protein distribution, such as various cytoplasmic organelles. Regarding the prediction accuracy metric IoU, CELL-Diff outperform CELL-E2 when using only the nucleus image as the conditional cell image, which can be largely improved by incorporating additional cell morphology images, such as those of the ER and microtubules.

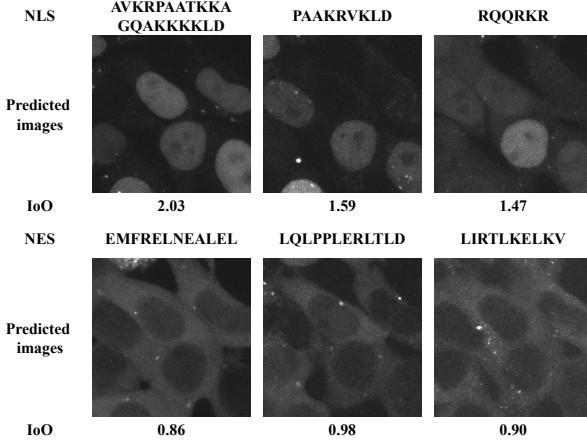

*Figure 6.* Protein localization signal screening. Test sequences are tagged to the C-terminus of the protein BLVRA. The ratio "IoO" represents the median protein intensity inside the nucleus relative to that outside the nucleus.

Regarding the learning-based metric FID, CELL-Diff significantly outperforms CELL-E2, further demonstrating the superiority of the proposed method. Visual results are illustrated in Figure 4. From the figure, we find that CELL-Diff accurately predicts protein images from unseen protein sequences. Compared with CELL-E2, CELL-Diff generates more resolvable images, enabling the extraction of more detailed information from the generated images. More results are provided in Appendix B.

## 6. Discussions

### 6.1. Ablation on cross attention module

We employ the cross-attention mechanism to more effectively integrate information from sequences to images. To evaluate its efficiency, we conduct an ablation analysis of

*Table 2.* Ablation analysis of cross attention module on HPA dataset. The nucleus image is used as the cell morphology image.

| Metric | w/o CA | **w CA** |
|---|---|---|
| MSFR (nm) $\downarrow$ | 687 | 641 |
| IoU $\uparrow$ | 0.434 | 0.484 |
| FID-T $\downarrow$ | 55.8 | 60.1 |
| FID-O $\downarrow$ | 41.3 | 51.1 |

this module on the HPA dataset, see Table 2. The results show that incorporating this module significantly improves the accuracy metric IoU, demonstrating the effectiveness of the cross-attention mechanism.

## 6.2. Potential applications

In this section, we present three potential applications of the proposed CELL-Diff method for biological discovery. Given that validation relies on biological knowledge and the dataset size is limited, we retrain all models using all the protein sequences from the HPA and OpenCell datasets.

**Virtual staining.** Typical fluorescence microscopes can only fit no more than four color channels in the visible spectrum. Because of this physical limitation, both HPA and OpenCell acquire the images of only one protein of interest per sample, with the other color channels occupied by morphological reference images. Consequently, it is challenging to identify the intracellular spatial relationships among multiple proteins of interest because their images are from different cells. With CELL-Diff, we solve this problem by generating images of these proteins conditioned on the same morphology reference images. These virtual staining images allow the subcellular distributions of an arbitrary number of proteins to be directly compared and potential molecular interactions identified from colocalization, while entirely circumventing the color channel limitation of fluorescence microscopy experiments. We demonstrate using CELL-Diff to identify molecular interaction by virtually staining two microtubule components (TUBA1A and TUBB4A) and two other proteins, KIF18A and EML4, from the same morphology image. The overlaid image clearly shows the association of KIF18A and EML4 with microtubules in the cell, consistent with their known biological function of microtubule binding, see Figure 5.

**Virtual screening of protein localization signal.** CELL-Diff can be applied for the virtual screening of protein localization signals, such as Nuclear Localization Signals (NLS) and Nuclear Export Signals (NES). The NLS is a short amino acid sequence that directs the import of proteins into the nucleus, while the NES facilitates their export from the nucleus. In this approach, the test peptide sequence is tagged to the C-terminus of the protein BLVRA, which is uniformly distributed both inside and outside the nucleus. CELL-Diff is then employed to predict the images of the modified protein. The resulting predicted images are analyzed to identify potential localization signals. As illustrated in Figure 6, we compute the median fluorescence intensity inside the nucleus relative to that outside the nucleus, referred to as the IoO ratio. For the original BLVRA protein, the IoO ratio is 1.12. If the IoO ratio of the modified protein exceeds 1.12, the test sequence is likely to function as an NLS, conversely, if the ratio is lower, the sequence is more likely to act as an NES. In Figure 6, we tested three NLSs and three NESs known from the literature. CELL-Diff successfully recognized these signals, proving its capability as a computational tool for screening potential protein localization signals.

**Localization signal generation.** Using image-to-sequence generation, CELL-Diff can be applied to generate novel protein localization signals, such as NLS and NES. Given a cell morphology image and a corresponding protein image, CELL-Diff generates the protein sequences that should be located at the position indicated by the protein image. Conditioned on an image of either a nucleus-localized protein or a nucleus-excluded protein (Figure 9), we generated 200 potential NLS and NES sequences, see Table 3 and Table 4, respectively. For NLS generation, the results indicate that the generated sequences exhibit key features of canonical NLS motifs, particularly clusters of basic residues (e.g., K-K/R-X-K/R). The sequences are enriched with lysine (K) and arginine (R) residues, characteristic of functional NLS motifs. Similarly, for the NES generation, the predicted sequences contain a high proportion of hydrophobic residues, including L, I, F, M, and V, which are known to be critical for NES function. The results suggest that CELL-Diff can effectively generate peptide sequences with potential localization signals, offering a promising tool for studying intracellular protein targeting.

## 7. Conclusion

This paper proposes CELL-Diff, a unified diffusion model that facilitates the transformation between protein sequences and microscopy images. Given cell morphology images as conditional inputs, CELL-Diff generates protein images from protein sequences. Conversely, it can generate protein sequences based on microscopy images. The objective function of CELL-Diff is constructed by integrating continuous and discrete diffusion models. Experimental results on the HPA and OpenCell datasets demonstrate that CELL-Diff produces more accurate results with higher resolvability than previous methods. Potential applications, including virtual screening of protein localization signals, virtual staining, and protein localization signal generation, make CELL-Diff a valuable tool for investigating subcellular protein localization and interactions.

## Impact Statement

This paper introduces a generative model that bridges protein sequences with their corresponding microscopy images, offering a novel tool for advancing biological research and discovery. The proposed approach has the potential to enhance our understanding of protein sequences and their subcellular localizations, paving the way for breakthroughs in molecular and cellular biology.

## Acknowledgements

We thank Chan Zuckerberg Biohub San Francisco Scientific Compute Platform for their support. We thank Guoxun Zhang for valuable discussions. This work is supported by the Chan Zuckerberg Biohub San Francisco Investigator gift to B.H..

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

# A. Implementation of Discrete Diffusion Model

The training and sampling process of the discrete diffusion model OA-ARDM (Hoogeboom et al., 2022) can be facilitated through a masking operation. Denote $\mathcal{C}$ as the categorical distribution, the training and sampling algorithms are shown in Algorithm 1 and Algorithm 2, respectively. For each training iteration, we first sample a time step $t$ from $\mathcal{U}(1, \ldots, D)$, and a random ordering $\sigma$ from $\mathcal{U}(S_D)$. Subsequently, we generate a mask $\mathbf{m}$ based on the index $i$ such that $\sigma(i) < t$. We then apply the network $\mathbf{f}_\theta$, which takes $\mathbf{m} \odot \mathbf{S}$ as input, and predicts the masked values $(1 - \mathbf{m}) \odot \mathbf{S}$.

---

**Algorithm 1** Training OA-ARDM

**Require:** Network $\mathbf{f}_\theta$, datapoint $\mathbf{S}$.
**Ensure:** $L_{\text{OA-ARDM}}$.
 1: Sample $t \sim \mathcal{U}(1, \ldots, D)$.
 2: Sample $\sigma \sim \mathcal{U}(S_D)$.
 3: Compute $\mathbf{m} \leftarrow (\sigma < t)$.
 4: Compute $\mathbf{l} \leftarrow -(1 - \mathbf{m}) \odot \log \mathcal{C}(\mathbf{S}|\mathbf{f}_\theta(\mathbf{m} \odot \mathbf{S}))$.
 5: $L_{\text{OA-ARDM}} \leftarrow \frac{1}{D-t+1}\text{sum}(\mathbf{l})$.

---

**Algorithm 2** Sampling from OA-ARDM

**Require:** Network $\mathbf{f}_\theta$.
**Ensure:** Sample $\mathbf{S}$.
 1: Initialize $\mathbf{S} = \mathbf{0}$, sample $\sigma \sim \mathcal{U}(S_D)$.
 2: **for** $t = 0, 1, 2, \ldots, D$ **do**
 3: $\quad \mathbf{m} \leftarrow (\sigma < t)$ and $\mathbf{n} \leftarrow (\sigma = t)$.
 4: $\quad \mathbf{S}' \sim \mathcal{C}(\mathbf{S}|\mathbf{f}_\theta(\mathbf{m} \odot \mathbf{S}))$.
 5: $\quad \mathbf{S} \leftarrow (1 - \mathbf{n}) \odot \mathbf{S} + \mathbf{n} \odot \mathbf{S}'$.
 6: **end for**

---

# B. Protein image generation

We present more protein image generation results. The results on the HPA and OpenCell datasets are shown in Figure 7 and Figure 8, respectively. From these results, we observe that CELL-Diff is capable of generating realistic protein images with high accuracy, enabling the discernment of fine details. Compared to CELL-E2, CELL-Diff produces images with higher resolvability, which provides better clarity of detailed localization structures.

# C. Localization signal generation

We use CELL-Diff to generate protein localization signals. Specifically, we use the images in Figure 9 as the conditional input for generating NLS and NES signals. Using the CELL-Diff model, we generate short amino acid sequences. A total of 100 potential sequences are generated for each signal type. Generated NLS and NES sequences are summarized in Table 3 and Table 4, respectively.

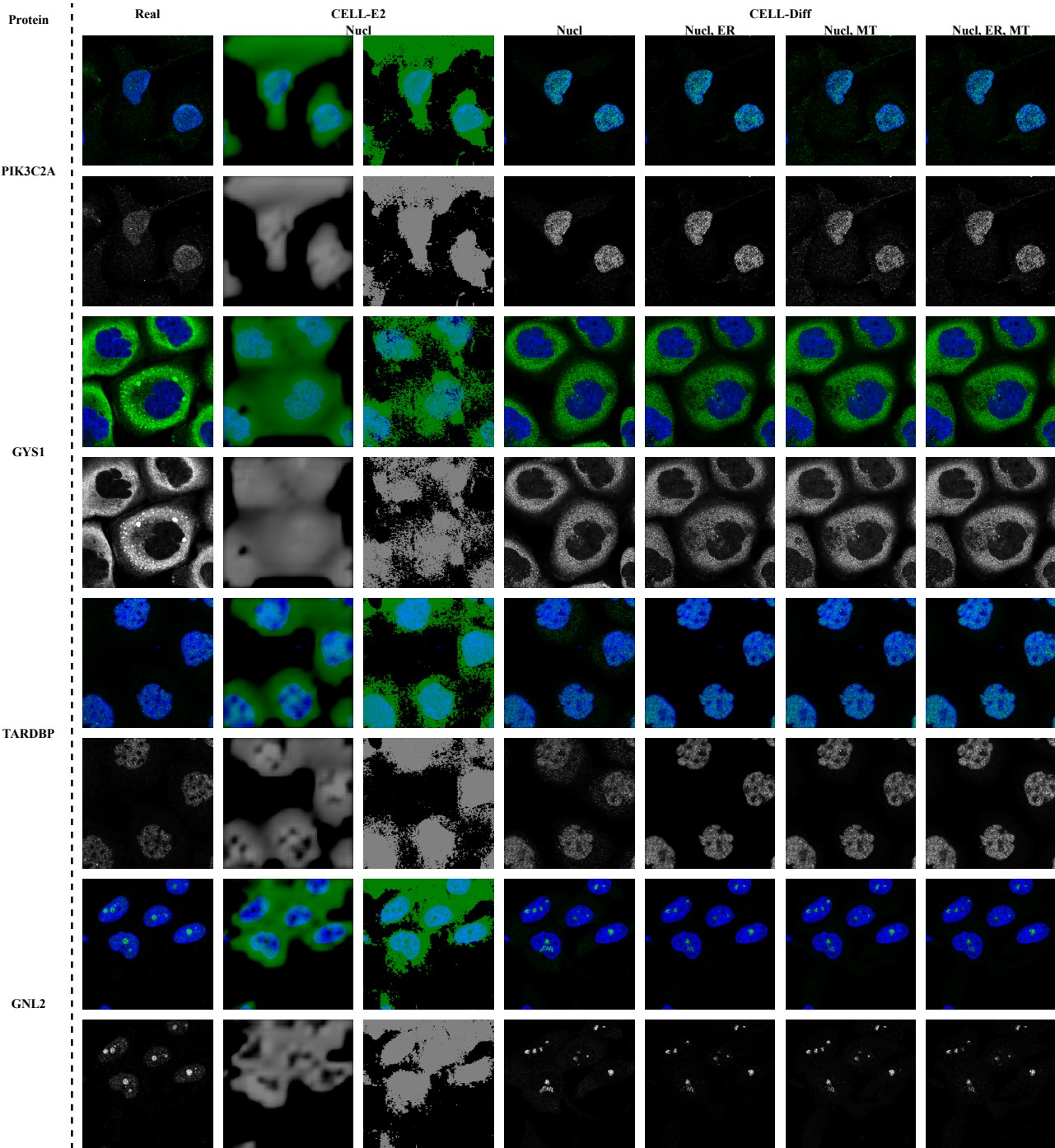

*Figure 7.* Visual results of protein image generation on HPA dataset.

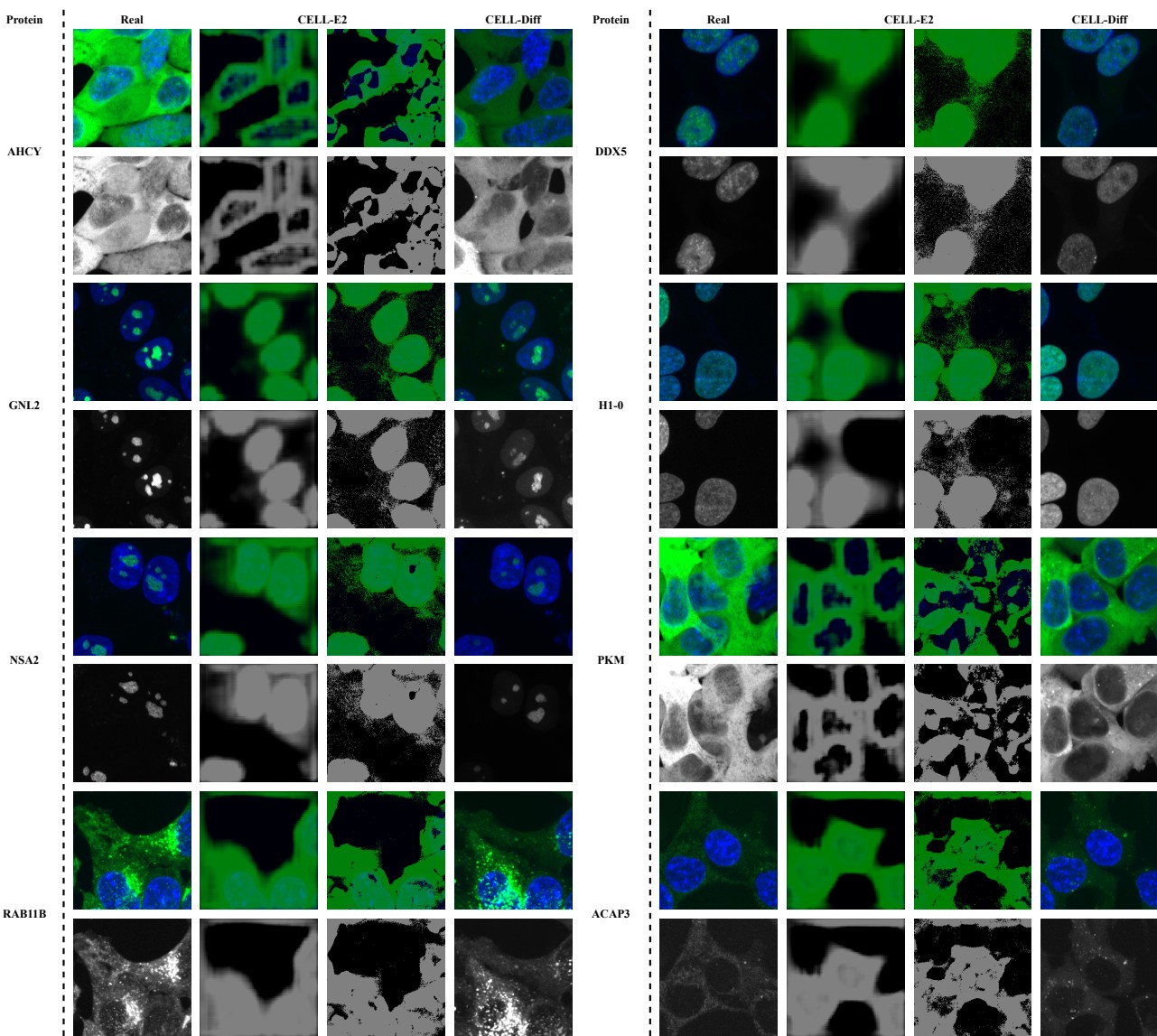

*Figure 8.* Visual results of protein image generation on OpenCell dataset.

*Table 3.* Generated NLS sequences.

| Index | Sequence | Index | Sequence |
|-------|----------|-------|----------|
| 1 | PERKK | 51 | PKRKAEGDAKGDKAK |
| 2 | PKGKK | 52 | SKRKAEGDAKGDKAKK |
| 3 | PKRSK | 53 | PKRKAEGDAKGDTAKK |
| 4 | PKRKK | 54 | PKRKAEGDAKGDKAKK |
| 5 | PEGKK | 55 | PKRKAPGDAKGDKAKKK |
| 6 | PKRKAE | 56 | PKRKAEGDAKGDKAKEK |
| 7 | PKGAKK | 57 | AKRKAKADAKGDTAKEK |
| 8 | PKRAKK | 58 | PKRKAEGDAKGDKAKKK |
| 9 | PKRAAK | 59 | SKRKAKNDAKGDKAKEKD |
| 10 | AKRKKK | 60 | PKRKAEGDAKGDKAKKKD |
| 11 | PKRKAK | 61 | PKRKKENDAKGDKAKKKD |
| 12 | PKRKAKK | 62 | PKRKAKADAKGETAKEKD |
| 13 | PKRKKKK | 63 | PKRKSPNDAKGPTTKKKD |
| 14 | PKGKAKK | 64 | PKRKKPADAKGAKAKKKD |
| 15 | PKRQAKK | 65 | PKRKAKNDAKGPTAKEKD |
| 16 | PKGKAKA | 66 | SKRKTPASAKGPKAKKKDG |
| 17 | PKRKAKAA | 67 | SKRKTVKDAKGDTAKKKDG |
| 18 | PKGKAKKA | 68 | SKRKTVASAKGPKAKKKDG |
| 19 | PKRKAKAD | 69 | SKRKAKADAKGETAKKKDG |
| 20 | PKRKAKGK | 70 | SKRKKEADAKGGDAKKKDG |
| 21 | PKRKAKGD | 71 | SKRKTPAPAKGEKAKKKDG |
| 22 | PKRKAKGA | 72 | PKRKTKAPAKGDKAKKKDE |
| 23 | PKRKAKEDA | 73 | PKRKTPAPAKGPTAKKKDG |
| 24 | PKRKAEKQA | 74 | SKRKTVASAKGPKAKKKDGP |
| 25 | PKRKAKADK | 75 | PKRKTKAPAKGEKTKKKDGP |
| 26 | PKRKAKRDA | 76 | KKRKPEADFKGDKAGKKDGPQ |
| 27 | PKRKAEADAK | 77 | PKRKAEADAKGDKAKKKDVPQ |
| 28 | PKRKAEASAK | 78 | SKRKAPADAKGDKAKKKDVPQ |
| 29 | PKRKKEAPAK | 79 | PKRKAPADAKGDKAKKKDEPQ |
| 30 | PKRKKPADAK | 80 | PKRKAEADAKGDKAKKKDVPQR |
| 31 | PKRKKEADAKG | 81 | PKRKAEKDAKGDKAKKKDEPQR |
| 32 | PKRKPEAPAKG | 82 | PKRKAPADAKGDKAKKKDVPQQ |
| 33 | PKRKKKAPAKG | 83 | PKRKAEADAKGDKAKKKDEPQR |
| 34 | PKRKSKASAKG | 84 | SKRKAPADAKGDKAKKKDVPQR |
| 35 | PKRKSEASAAGE | 85 | PKRKAEADKKGDKAKKKDEPQR |
| 36 | PKGKTEADAAGG | 86 | SKRKAPADAKGDKAKKKDMPQQ |
| 37 | PKRKAPAPAKGD | 87 | AKRKAEADAKGDKAKKKDEPQQR |
| 38 | PKRKKEAPAKGE | 88 | PKRKAEGDAKGDKAKKKDEPQRR |
| 39 | PKRKAPAPAAGD | 89 | PKRKAPADAKGDKAKKKDGPQQR |
| 40 | PKRKSKASAEGG | 90 | AKRKAERDAKGDKAKEKDEPQQAG |
| 41 | PKRKAEEDAKGAK | 91 | AKRKAERDAKGDKAKVKDEPQRAG |
| 42 | PKRKAEAAAKGAG | 92 | PKRKAERDAKGDKAKKKDEPQQAG |
| 43 | PKRKAEADAKGET | 93 | AKRKAERDAAGDKAKKKDEPQQAG |
| 44 | PKRKAEADAAGEK | 94 | AKPKAPRDAKGDKAKKKDEPQQAG |
| 45 | PKRKAEADAPGDG | 95 | AKRKAEADAKGDTAKEKDEPQQAGA |
| 46 | PKRKAEEDAKGDKA | 96 | PKRKPEADAKKDTAKEKDEPQQAGA |
| 47 | PKSKAEEDRKGPKA | 97 | AKPKAEKDAKKDKAKEKDEPQQAGA |
| 48 | PKRKAEEDAKGPKA | 98 | AKRKAERDAKGETAKEKDEPQQAGA |
| 49 | PKGKAEADAKGDKA | 99 | PKRKAERDAKGDTAKKKDEPQQAGA |
| 50 | PKRKAEEDAKGDKAK | 100 | SKRKAETDAKGDTAKEKDEPQQAGA |

Table 4. Generated NES sequences.

| Index | Sequence | Index | Sequence |
|---|---|---|---|
| 1 | AMFKALREMGG | 51 | AMFAARRLLGKAAQLDY |
| 2 | QDIIARIEQGK | 52 | VSFSARLLLGFMAQLDT |
| 3 | GMIRARRENGE | 53 | AMFAARLLLGEFAQRDY |
| 4 | QMIIVLREMKG | 54 | MMKIARLLLGKMAQLDT |
| 5 | AMIRVRIEMGE | 55 | AGFIARLLLGFFAQLDY |
| 6 | ADIIALREMGKA | 56 | AMFIARLLHGFMAQLDY |
| 7 | QKARAPEGELGA | 57 | AMIIARRELDFFAQLDTF |
| 8 | ADIIAAEEVGKA | 58 | AMIIARRELGKFAQYDTF |
| 9 | ADIIAAEEVIKA | 59 | AMFIARRSLGEFAQLDTF |
| 10 | ADIIAPRLHGKA | 60 | AMIIARRSLGKFAQYDTD |
| 11 | ADIIAPEEMGKE | 61 | AMFIARRSLGKFAQLDTF |
| 12 | GDIIAPEEYGKA | 62 | AMIIARRELQFFQLDTF |
| 13 | ADIIAPRKVIKA | 63 | AMISARRENGKFAQKDTFL |
| 14 | ADIIARDEMGKA | 64 | AMIIARREHGFFFQKDRFC |
| 15 | ADIIAPEEVGKA | 65 | AMIIARIEHGFFAQLDLFQ |
| 16 | GDFLAARKKGGFF | 66 | AMIIARKLNGKFAVLDQFN |
| 17 | GDFLAPEKVGEFF | 67 | AMISARDENYVFAALDQFP |
| 18 | GDQLAPLKYGKFF | 68 | AMISARREHNKFAAIDTFF |
| 19 | CDIEAPEKLPRFF | 69 | AMISARREHQKFALIDTFC |
| 20 | GDDLAPLRVGKFF | 70 | AMISARRKYLKFAQIDTFF |
| 21 | CDIRAAEQVGKFE | 71 | AMIIARRELTGFGADDTFF |
| 22 | SDILAPRKHGEFF | 72 | AMISARREYDKFAQLDQFQ |
| 23 | PDKLAPLEVGGFFR | 73 | AMISALIEHGFFAVCDLKFR |
| 24 | KDKKAPRQVGGFFL | 74 | AMIIARKEHTVFAEIDQFFR |
| 25 | GDFIAPQLFGGFFR | 75 | ADIIALIKHTFFELIDLFQR |
| 26 | SDFLAPELFGGFFE | 76 | SMIIARRENGFFAQIDLFFR |
| 27 | PDFKAPQLVKGFFA | 77 | ALIIALIENQGFEVIDQKFR |
| 28 | PDFLVPLEFGEFFA | 78 | ALIIARKEHPFFAIIGLFFR |
| 29 | PDFLAPLLVPLFFA | 79 | ADISAVIEHKFFAVIDLFFL |
| 30 | PGFIAEQLKGLFAQE | 80 | AQIIALRENPVFAQSGLFFR |
| 31 | PDFEAPQLKLGFAEL | 81 | GLISALIEHPGFFALDQFFR |
| 32 | CDFLVDLLLVLFAAI | 82 | AQIIALIEHDGFATLDQDFR |
| 33 | PGKEAPQLLPKFAAI | 83 | GDIAALIKFQKFAVIGLNQRV |
| 34 | PGFEANQLFGEFFAY | 84 | SDFIALISFQGFAVLGLFQLV |
| 35 | PDFEAPQLQLGFATI | 85 | SDIRALLRNDGFAALGLKQPE |
| 36 | PDFEAELLKLLFAIE | 86 | SDIAALIRKGGFEAIGQFFRC |
| 37 | PGFLADLLHLLFQAV | 87 | ADIRALIKNKKFAILGQKQPE |
| 38 | PDFEAPQLFGKMAAK | 88 | GDKAALKEFGGFEQLGLFQPE |
| 39 | PDFEANQLKLGFAEV | 89 | SDIAALIEFPGFEALDLPQPV |
| 40 | VGFSARQLHGKFAQVI | 90 | GDIRALIKFQKFEVIGQKQPE |
| 41 | SGFSARQLKGKMAQLI | 91 | SDISALIRNHGVADLGQFQRC |
| 42 | AGFSAEQLLGKFAQEI | 92 | GDISALISNKGFAEIGQFQSC |
| 43 | AGFAADQLLGKMAQLI | 93 | CKKLALLSFVGFAVCDTKQSED |
| 44 | MGFSADQLLGKMAQLI | 94 | CKKRALDRNKFMAALLLKQAKD |
| 45 | SGFSARQLQGKFAEED | 95 | SDIRALKKFQKFAVLGLGQLFD |
| 46 | AGFSADQLLGKFAQLI | 96 | SKKRALDKFVGFAVLGLFLRVD |
| 47 | AGFLADQLLGKFAQVI | 97 | GMKRALDKFLGFAALGTFQPFD |
| 48 | AGFSADQLLGKMAQLI | 98 | AMKRALDSFQKFAASLLKQEED |
| 49 | PGFSAEELLGKKAQVI | 99 | AKKIALDSNKGFAALLLFQVED |
| 50 | MMFSARLLLQEFAQEDR | 100 | AKKRALDKFLGFSVLDLFQEED |

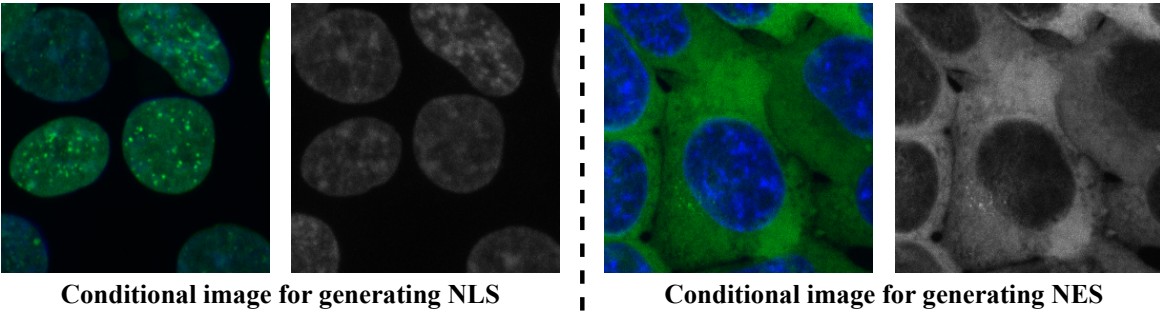

**Conditional image for generating NLS**          **Conditional image for generating NES**

*Figure 9.* Conditional images for protein localization signal generation.

