# OpenReview forum: "Bridging Protein Sequences and Microscopy Images with Unified Diffusion Models"
_ICML.cc/2025/Conference — ICML 2025 poster_

### Official Review · Reviewer_nmZP · 2025-03-13

**Overall Recommendation:** 3

**Summary:**

This paper introduces CELL-Diff, a diffusion-based model capable of bidirectionally generating microscopy images from protein sequences and protein sequences from microscopy images. Using conditional morphology reference images (nucleus, ER, microtubules), the model combines continuous diffusion for images and discrete diffusion for sequences. CELL-Diff significantly improves image quality compared to previous methods, evaluated on the HPA and OpenCell datasets.

**Claims And Evidence:**

The authors claim CELL-Diff provides high-fidelity microscopy image generation and accurate sequence-to-image and image-to-sequence transformations. While the image generation performance is strongly supported through quantitative (FID, IoU, MSF metrics) and qualitative evaluations, the claim of accurate image-to-sequence transformation lacks rigorous quantitative evidence. Evidence provided for sequence generation is limited to qualitative motif analyses.

**Essential References Not Discussed:**

N/A

**Experimental Designs Or Analyses:**

The experimental design for image generation is solid with clear metrics and appropriate baselines.

**Methods And Evaluation Criteria:**

- The methodological choice (combining continuous diffusion for images and discrete diffusion for sequences in a unified transformer-based U-Net) is interesting and sound.
- However, the authors' evaluation of image-to-sequence generation is weak, relying primarily on qualitative assessments rather than robust quantitative analyses. The paper could benefit from a larger-scale motif analysis / cluster overlap with localization etc.

**Other Comments Or Suggestions:**

N/A

**Other Strengths And Weaknesses:**

N/A

**Questions For Authors:**

- Can you please share your thoughts on previous sections?
- How deterministic or diverse are the sequences generated from a given microscopy image?
- Is there any cycle consistency between the two directions? Say for a protein sequence, generate an image, and then feed that generated image back into the model to generate a sequence, do you get back similar sequence?

**Relation To Broader Scientific Literature:**

The paper adequately discusses relevant works in the field.

**Theoretical Claims:**

N/A

---

> ### Author Rebuttal · Authors · 2025-04-01
>
> **Comment 1:** The claim of accurate image-to-sequence transformation lacks rigorous quantitative evidence. Evidence provided for sequence generation is limited to qualitative motif analyses.
>
> **Response:** To quantitatively evaluate the accuracy of the generated sequences, we used DeepLoc 2.1[1] to assess whether the generated sequences contain recognizable Nuclear Localization Signals (NLS) motifs. DeepLoc 2.1 is a deep learning-based classification model for predicting protein subcellular localization based on discrete annotations.
>
> Specifically, we fused the generated sequences to the C-terminus of Green Fluorescent Protein (GFP) and input the fused proteins into DeepLoc 2.1. The model then predicted whether the fusion proteins contained NLS.
>
> We compared CELL-Diff's performance with CELL-E2. For CELL-Diff, we used the 100 generated proteins listed in Table 3, while for CELL-E2, we evaluated the first 100 sequences from their supplementary materials, ranked by their generation scores.
>
> According to DeepLoc 2.1, 78 out of 100 CELL-Diff-generated sequences were predicted to contain NLS motifs, while only 46 out of 100 CELL-E2-generated sequences were recognized as NLS motifs. This indicates that CELL-Diff has a higher probability of generating NLS-containing sequences compared to CELL-E2.
>
> ---
>
> **Comment 2:** How deterministic or diverse are the sequences generated from a given microscopy image?
>
> **Response:** To assess the diversity of sequences generated from a given microscopy image, we performed a diversity analysis on 100 generated NLS sequences. We used the following three metrics: Levenshtein Distance, Sequence Entropy, and Tanimoto Diversity. The results are shown below:
>
> | **Model**    | **Levenshtein Distance** | **Sequence Entropy** | **Tanimoto Diversity** |
> |--------------|---------------------------|-----------------------|------------------------|
> | CELL-E2      | 14.25                     | 3.83                  | 0.99                   |
> | CELL-Diff    | 9.91                      | 2.87                  | 0.83                   |
>
> From the results, we observe that CELL-E2 produces more diverse NLS sequences compared to CELL-Diff. However, the lower diversity in CELL-Diff reflects its stronger conditioning on the input image, which leads to more biologically meaningful and consistent sequence patterns. As demonstrated in response to Comment 1, CELL-Diff generates NLS sequences more likely to be valid, as confirmed by DeepLoc 2.1 analysis.
>
> ---
>
> **Comment 3:** Is there any cycle consistency between the two directions? Say for a protein sequence, generate an image, and then feed that generated image back into the model to generate a sequence, do you get back a similar sequence?
>
> **Response:** In general, the space of protein sequences is much larger than the functional space of images. For instance, proteins with different NLSs can produce visually indistinguishable images, as they all localize to the nucleus. Therefore, if we start with one NLS sequence, generate an image showing nuclear localization, and then use that image to generate a new sequence, the resulting sequence will likely contain a different NLS but still direct the protein to the nucleus. Hence, sequence-to-image-to-sequence cycle consistency is not a suitable measure of the model's performance, as it does not guarantee the recovery of the same sequence.
>
> However, image-to-sequence-to-image cycle consistency is more meaningful, as it evaluates whether the generated sequence preserves the same localization or pattern as the original input image. To assess this, we conducted a cycle consistency validation experiment using NLS localization.
>
> We started with a protein image showing nuclear localization, similar to Figure 9 (left). We then used the model to generate 10 sequences fused to the C-terminus of GFP with different sequence lengths. Next, we fed the generated sequences back into the model to generate corresponding protein images.
>
> To quantify the consistency, we measured the IoU similarity between the original and regenerated images. We compared CELL-Diff with CELL-E2, and the results are presented below:
>
> | **Number of Amino Acids** | **CELL-E2** | **CELL-Diff** |
> |----------------------------|-------------|----------------|
> | 10                         | 0.575       | 0.763          |
> | 20                         | 0.566       | 0.753          |
> | 30                         | 0.579       | 0.764          |
> | 40                         | 0.574       | 0.751          |
>
> From the results, CELL-Diff demonstrates better cycle consistency than CELL-E2, as indicated by the consistently higher IoU scores. This suggests that CELL-Diff generates sequences that better preserve the localization and pattern information of the original input image.
>
> ---
>
> [1]: Ødum, Marius Thrane, et al. "DeepLoc 2.1: multi-label membrane protein type prediction using protein language models." *Nucleic Acids Research* 52.W1 (2024): W215-W220.

---

### Official Review · Reviewer_sVh2 · 2025-03-14

**Overall Recommendation:** 4

**Summary:**

The paper introduces CELL-Diff, a unified diffusion model designed for bidirectional transformations between protein sequences and their corresponding microscopy images.  Given cell morphology images and a protein sequence, CELL-Diff generates corresponding protein images.  Conversely, it can also output protein sequences from protein images.  CELL-Diff integrates continuous and discrete diffusion models within a unified framework and is implemented using a transformer-based network.  The model is trained on the Human Protein Atlas (HPA) dataset and fine-tuned on the Open-Cell dataset.  Experimental results demonstrate that CELL-Diff outperforms existing methods in generating high-fidelity protein images.

**Claims And Evidence:**

Yes
1.Claim: CELL-Diff facilitates bidirectional generation between protein sequences and images.

1.Evidence: The paper provides Figure 1 and discusses the model's ability to generate protein images from sequences and vice versa.  The methodology section details how the model is trained to handle both types of generation.  Experimental results in Section 5.3 and Appendix B also visually support this claim.

The claim is supported by the presented evidence.

2.Claim: CELL-Diff outperforms existing methods in generating high-fidelity protein images.

2.Evidence: The paper compares CELL-Diff with CELL-E2 and uses metrics like MSF-resolvability, IoU, and FID to demonstrate superior performance.  Visual comparisons in Figure 4 and Appendix B also highlight the improved image quality.

The claim is well-supported by quantitative and qualitative evidence.

3.Claim: CELL-Diff can be applied for virtual screening of protein localization signals, virtual staining, and protein localization signal generation.

3.Evidence: Section 6.2 details these potential applications and provides supporting figures (Figure 5, 6, and 9) and generated sequence tables.

The applications are well-described, and the results seem promising, but further experimental validation would strengthen these claims.

**Essential References Not Discussed:**

Some reference of Stable Diffusion

**Experimental Designs Or Analyses:**

The experiments are designed to evaluate the protein image generation performance of CELL-Diff and to demonstrate its potential applications.

The model is trained on the HPA dataset and fine-tuned on the OpenCell dataset.

The performance is compared with CELL-E2 using MSF-resolvability, IoU, and FID metrics.

Ablation studies are conducted to evaluate the effectiveness of the cross-attention mechanism.

Potential applications are demonstrated through virtual staining, virtual screening of protein localization signals, and localization signal generation.

**Methods And Evaluation Criteria:**

Methods: The proposed CELL-Diff method combines continuous and discrete diffusion models within a unified framework.  It employs a transformer-based U-Net architecture with cross-attention mechanisms.  The training objective function includes noise prediction loss for the continuous diffusion model and masked value prediction loss for the discrete diffusion model.  A latent diffusion model is used to reduce computational costs.

The methods are clearly described and seem appropriate for the problem. The combination of continuous and discrete diffusion, along with the transformer-based architecture, is a reasonable approach.

Evaluation Criteria: The paper uses metrics:

1.MSF-resolvability: This metric measures the capability to discern fine structural details in microscopy images.

2.IoU (Intersection over Union): This metric measures the similarity between two masks, used here to compare predicted and real protein image masks.

3.FID (Fréchet Inception Distance): This metric evaluates the similarity between the real and generated images regarding their feature distributions.

These evaluation criteria are appropriate for assessing the quality and accuracy of generated microscopy images. MSF-resolvability is a particularly relevant metric for this task.

**Other Comments Or Suggestions:**

In the methodology section, providing more visual aids or diagrams to illustrate the diffusion processes and the network architecture could further improve clarity.

It would be beneficial to discuss the computational cost and scalability of CELL-Diff in more detail, as this is an important consideration for practical applications.

**Other Strengths And Weaknesses:**

Strengths:
1.The paper addresses an important problem: understanding the relationship between protein sequences and their cellular functions.
2.The proposed model has the potential to be a valuable tool for investigating subcellular protein localization and interactions, with potential applications in drug discovery and disease research.
3.The paper demonstrates the model's potential through several applications, including virtual staining, virtual screening of protein localization signals, and localization signal generation.

Weakness:
1.While generally clear, some parts of the paper, especially the technical details of the model and the training process, could be challenging for readers without a strong background in machine learning and diffusion models.
2.The datasets used in the experiments, while relevant, might be considered limited in size and diversity, which could affect the generalizability of the model.

**Questions For Authors:**

The authors mentioned that "Some parts of the paper, especially the technical details of the model and the training process, could be challenging for readers without a strong background in machine learning and diffusion models." Could the authors provide more clarification or additional explanations on these technical details to make the paper more accessible to a broader audience?

The authors used the HPA and OpenCell datasets for their experiments. Given the limited size and diversity of these datasets, could the authors discuss the potential impact of this limitation on the generalizability of the model and how future work could address this?

**Relation To Broader Scientific Literature:**

The paper builds upon previous work in the field of predicting protein properties using learning-based methods. It cites examples such as predicting protein structure, interaction partners, and subcellular localization.

It is related to the development of generative models for designing functional proteins and drug-like molecules.

The work focuses on the relationship between protein sequences and their cellular functions, as characterized by microscopy images, particularly fluorescence microscopy.

It is specifically related to recent work that proposed CELL-E, a text-to-image transformer that predicts fluorescence protein images from sequence input and cell morphology condition, and its enhancement CELL-E2.

**Theoretical Claims:**

The paper includes theoretical claims related to the diffusion models and the derivation of the training objective.

1.The forward process of the continuous diffusion model is defined in Equation 1, and the subsequent derivation of  q(I_t∣I_0) is standard.

2.The reverse process is defined in Equation 2, which is also a standard formulation.

3.The ELBO (Equation 3) and its simplified form (Equation 4) are correctly presented.

4.The derivation of the loss function for OA-ARDM (Equation 7) appears to be correct.

5.The conditional ELBO for continuous and discrete diffusion models (Equations 8 and 9) are derived logically from the previous equations.

6.The combined loss function (Equation 12) is a straightforward combination of the individual losses.

---

> ### Author Rebuttal · Authors · 2025-04-01
>
> Due to the limitation on the number of characters, we provide a general response to the reviewer's comment.
>
> ### 1. The dataset size and model generalizability.
>
> The generalizability of the sequence-to-image task depends on the downstream application.
>
> One application is **virtual staining**. Traditional fluorescence microscopy images typically have four color channels, limiting the ability to visualize the spatial relationships between multiple proteins of interest. With CELL-Diff, we can virtually stain all proteins from the training dataset using the same cell morphology images. As shown in Figure 5, this enables the identification of potential protein-protein interactions. In this context, the generalizability pertains to the conditional cell morphology images. Since both the HPA and OpenCell datasets contain approximately 100 cells for each protein, the model can learn diverse cellular morphologies, achieving strong generalizability with respect to different cell morphologies.
>
> Another application is **sequence-to-image prediction**, which involves generating images for unseen proteins. In cell biology, rather than predicting images for entirely artificial sequences, the focus is often on generating images for biologically relevant variants, such as protein truncations or point mutations. Consequently, the effective protein sequence space in practical applications is much smaller than in tasks like de novo protein design or unconditional sequence generation.
>
> For specific biological problems, such as predicting protein phase separation or identifying disease-related mutations, task-specific datasets can be collected. These datasets are often easier to obtain compared to large-scale, general-purpose datasets. By fine-tuning CELL-Diff on task-specific datasets, the model's generalizability can be improved, making it adaptable to new biological contexts.
>
> To further validate the generalizability of CELL-Diff, we expanded the NLS screening experiment to the INSP yeast NLS dataset[1], which contains 50 valid NLSs from yeast. We generated new proteins by fusing the yeast NLS sequences to the C-terminus of Green Fluorescent Protein (GFP), which is not included in the HPA or OpenCell datasets, and tested CELL-Diff's ability to predict the function of these yeast NLSs.
>
> For the generated images, we quantified the median fluorescence intensity inside and outside the nucleus. If the median intensity inside the nucleus was higher than outside, we considered the model to have made a correct prediction. The total number of correct predictions is denoted as $N_{hit}$, and we computed the identification rate as $N_{hit}/N$, where $N$ is the total number of NLS-tagged proteins.
>
> To further evaluate the model's biological reasoning, we tested different fusion patterns by increasing the number of NLS tags. In practice, proteins fused with multiple NLS sequences have a higher efficiency of entering the nucleus. Thus, we evaluated three configurations: GFP + NLS, GFP + NLS + NLS, and GFP + NLS + NLS + NLS, which correspond to the increasing effectiveness of nuclear localization.
>
> We compared CELL-Diff's identification rate against CELL-E2, and the results are presented below.
>
> | **Test Protein**         | **CELL-Diff** | **CELL-E2** |
> |-------------------------|---------------|-------------|
> | GFP + NLS               | 0.82          | 0.54        |
> | GFP + NLS + NLS         | 0.86          | 0.78        |
> | GFP + NLS + NLS + NLS   | 0.90          | 0.88        |
>
> From the table, we observe that both CELL-Diff and CELL-E2 can effectively identify NLS-tagged proteins, with CELL-Diff demonstrating higher identification rates, especially for single NLS fusions. Additionally, the identification rate increases as more NLS tags are added, which aligns with real-world biological behavior. This experiment suggests that CELL-Diff captures some underlying biological logic rather than merely memorizing dataset-specific patterns.
>
> ### 2. Technical details and visual aids.
> We will include more technical details and visual illustrations of our method in the revision.
>
> ### 3. Computational cost and scalability.
> The CELL-Diff model contains approximately 1 billion parameters, while the VAE used for latent diffusion has 42 million parameters. The model was trained on 2 NVIDIA H200 GPUs for approximately 10 days. CELL-Diff is scalable due to its latent diffusion framework, which reduces computational costs by operating in a lower-dimensional latent space. This enables the efficient generation of high-resolution images. Furthermore, CELL-Diff can be scaled by increasing the model size and utilizing larger computational clusters, making it adaptable for more complex biological datasets and tasks.
>
> [1] Guo, Yun, et al. "Discovering nuclear targeting signal sequence through protein language learning and multivariate analysis."

---

### Official Review · Reviewer_igYh · 2025-03-14

**Overall Recommendation:** 1

**Summary:**

This paper introduces CELL-Diff, a unified diffusion model that enables bidirectional generation between protein sequences and fluorescence microscopy images. It combines continuous diffusion for image synthesis and discrete diffusion for sequence prediction, integrating transformer-based cross-attention to fuse multimodal representations. The model is trained on Human Protein Atlas and OpenCell datasets, demonstrating improvements in protein localization prediction and potential applications in virtual staining and protein sequence design.

## Update after rebuttal:
I do not see sufficient justification to change my original rating. Please see my reply to rebuttal below.

**Claims And Evidence:**

- The paper claims that CELL-Diff, a unified diffusion model, can bidirectionally transform between protein sequences and their corresponding fluorescence microscopy images by integrating continuous diffusion (for images) and discrete diffusion (for sequences) within a single framework.

- Quantitative metrics show better image generation than prior models. The concept of bidirectional transformation is novel in this context. Case studies suggest potential biological applications.

- Unseen proteins were not rigorously tested. CELL-Diff is most likely memorizing dataset-specific patterns. The paper oversells its claims. While diffusion models are powerful, they cannot overcome the fundamental limitation of tiny training datasets in an enormous search space. This is likely more of an interpolation method rather than a true sequence-to-image generative model.

**Essential References Not Discussed:**

These are not essential but literature review would be more complete with some of these diffusion-based structure prediction methods.

AlphaFold3 (Abramson et al., 2024) - AlphaFold that uses diffusion
RFdiffusion (Watson et al., 2023) – The first diffusion model for protein structure generation.
FrameDiff (Wu et al., 2023) – Rigid-body diffusion for protein structure prediction.
ProteinSGM (Trippe et al., 2022) – Diffusion for protein sequence design.
FoldFlow (Anand et al., 2022) – Normalizing flows for protein backbone generation.

**Experimental Designs Or Analyses:**

- The authors compare CELL-Diff to CELL-E2, a previous protein-to-image generation model. Cell-Diff improves image clarity over CELL-E2. Although visual inspection shows more detailed subcellular structures, none of the metrics used for comparison would necessarily correlate with biological accuracy.
- The image generation results seem promising within the dataset, but there is no proof of generalization to novel proteins. There is no held-out test set of completely unseen protein families, making it impossible to evaluate generalization. The sequence prediction claim is not validated by experts or in wet-lab studies. The methodological contribution (unified diffusion) is not rigorously tested. Molecular interaction predictions could be misleading. If two proteins often appear together in the dataset, the model might just learn their cooccurrence rather than discovering true interactions.
- While ambitious in scope, this paper attempts to tackle an inherently intractable problem without the necessary scale of data, established benchmarks, or biological validation, making its claims more speculative than substantive.

**Methods And Evaluation Criteria:**

- CELL-Diff conditions image generation on reference cell morphology images (e.g., nucleus, ER, microtubules). Uses latent diffusion models to process microscopy images efficiently. The training objective combines: Noise prediction loss for continuous diffusion (image generation). Masked value prediction loss for discrete diffusion (sequence prediction).

- Method uses several metrics that measures image clarity and fine structural details (MSF), assesses how well the generated protein images align with ground truth (IoU), evaluates similarity between generated and real images (Frechet Inception Distance)

**Other Comments Or Suggestions:**

n/a

**Other Strengths And Weaknesses:**

Strengths:
- The paper introduces an ambitious multimodal diffusion framework that, if validated, could open new directions in bridging protein sequences and cellular imaging

Weaknesses:
- The Training data is woefully small: The Human Protein Atlas (HPA) dataset contains around 10K proteins with fluorescence microscopy images. Even though prominent structure prediction methods also use hundreds of thousands of protein structures protein folding problem is not random meaning the number of viable structures is vastly smaller than the full combinatorial space. On the other hand microscopy images have no equivalent constraints. The mapping from sequence to fluorescence image is far less structured than sequence-to-structure. The same protein can exhibit multiple localizations depending on cell type, environment, modifications. No universal, physics-driven constraints like those in protein folding.
- Microscopy Images Are Not Sufficient to Capture Protein Function: Protein function is not solely determined by sequence; it depends on post-translational modifications, cellular environment, and binding interactions. Even if a model memorized all available images, it would not generalize to unseen proteins effectively.
- Bidirectional Mapping Between Sequence and Image is Ill-Defined: A single protein sequence can adopt multiple conformations and localizations depending on: Cell type, Post-translational modifications, Interaction partners. One-to-one mapping between protein sequences and fluorescence images does not exist.
- Diffusion Models May Be Overfitting: The authors claim CELL-Diff outperforms prior methods like CELL-E2, but if the model is trained on such a small dataset, it could be: Memorizing protein localizations rather than learning meaningful structure-function relationships. Hallucinating plausible but incorrect images, which may still look visually appealing but lack biological relevance.
- No Evidence of Generalization to Unseen Proteins: A true sequence-to-image model should be tested on novel sequences never seen in training, but the paper does not provide convincing results for this. Without proper benchmarking on completely held-out protein families, this model might just be fitting noise or dataset-specific patterns.

**Questions For Authors:**

n/a

**Relation To Broader Scientific Literature:**

- The paper positions itself at the intersection of protein sequence modeling, fluorescence microscopy, and generative AI. Although the scope might be slightly different, there are many methods that use diffusion to predict structure from sequence (AlphaFold3, RoseTTAFold, Baek et al. 2021, RFDiffusion Waston et al., 2023, FrameDiff Wu et al. 2023). RFdiffusion and FrameDiff use diffusion models, but they generate structured, atomic-level representations, whereas CELL-Diff tries to map sequences directly to microscopy images. There are also diffusion-based generative models for de novo protein sequence design.

- While ambitious, structure prediction and protein design have strong theoretical foundations and are being tackled by leading labs with rigorous validation, making them credible scientific pursuits. In contrast, sequence-to-microscopy image generation lacks structural constraints, relies on limited data, and has no established evaluation metrics, making it far more speculative. Moreover, the state space of high-resolution microscopy images is vastly larger than that of 3D protein structures, further highlighting the impracticality of learning a direct mapping from sequence to image.

**Theoretical Claims:**

- The paper does not have any strong theoretical claims. Section 3.1 presents the standard formulation of continuous diffusion models. Section 3.2 discusses OA-ARDM for discrete data (Hoogeboom et al. 2022). Uses random ordering to make the model agnostic to token positions.
- Section 4 of the paper introduces the CELL-Diff model and describes how it integrates continuous and discrete diffusion models for bidirectional transformation between protein sequences and microscopy images. The authors claim to integrate continuous diffusion and discrete diffusion within a single framework. Continuous diffusion follows standard diffusion formulations. Discrete diffusion follows OA-ARDM (Hoogeboom et al., 2022). Not a novel theoretical contribution. The loss formulations are also standard for diffusion models and masked language modeling.

---

> ### Author Rebuttal · Authors · 2025-04-01
>
> Due to character limitation, we provide a general response to the comments.
>
> We understand the reviewer’s concerns about the concept of sequence-to-image mapping, especially with limited data from HPA and OpenCell. While a universal sequence-to-image model requires massive data, CELL-Diff, as a smaller model, can still make practically relevant predictions for general cell biology. Specifically:
> ### 1. Feasibility of sequence-to-image mapping.
> While the subcellular localization of a protein is influenced by cell type, state, and post-translational modifications, many proteins have a defined localization, often determined by sequence motifs (e.g., NLS for nuclear localization, signal peptide for ER translocation, and CAAX motif for lipid modification and subsequent plasma membrane localization). This premise underpins localization prediction models like DeepLoc[1] and MultiLoc[2], and CELL-Diff extends this concept by using images rather than discrete annotations, allowing quantitative description of multi-localization. For example, Figure 6 shows that CELL-Diff can characterize the non-binary effectiveness of different NLSs and NESs using the nuclear-to-cytoplasmic signal ratio from segmented images. The importance of understanding this quantitative effectiveness is illustrated by the need to add three separate NLSs to Cas9 for sufficient nuclear localization in CRISPR/Cas9 genome editing. Moreover, the input condition image implicitly conveys information about cell type and state. For instance, the CELL-E paper demonstrated the ability to predict the spherical shape of a cytoplasmic protein in a mitotic cell using a DNA-stain condition image.
> ### 2. The dimension of the image space.
> Practical cell biology research often focuses on high-level image features like morphology and colocalization, rather than pixel-by-pixel data. In this sense, the meaningful image state space is likely smaller than the atomic coordinate structural space. As an illustration of this point, previous work (cytoSelf[3]) demonstrated the correlation between dimension-reduced image representations and stoichiometric protein-protein interactions and identified a new component of a protein complex solely by image similarity.
> ### 3. Dataset size and the validation of sequence-to-image prediction.
> Although CELL-Diff's training data is smaller than that for structure prediction models, the HPA dataset covers over 60% of human proteins. Combined with ESM2 embeddings, CELL-Diff makes meaningful predictions. Figure 6 shows accurate localization predictions for NLS- and NES-fused biliverdin reductases, including the KLKIKRPVK sequence from E. coli protein Tus, acting as a mammalian NLS. We also tested GFP (from jellyfish and non-homologous to human proteins) fused to various yeast NLSs and demonstrated the additive effect of multiple NLSs. These results showcase CELL-Diff's ability to extract biological knowledge about nuclear importin recognition from limited data (Reviewer sVh2, 1. The Dataset Size and Model Generalizability.).
>
> We must state that CELL-Diff is not a "true sequence-to-image generative model" capable of translating any random sequence into pixel-perfect cellular images, as structure prediction models do. Instead, it is built as a virtual experiment tool for typical sequence variabilities in cell biology research (mostly mutations of endogenous proteins). We will revise the manuscript to clarify its application and limitations.
> ### 4. Comparison with CELL-E2.
> The intention for us to develop CELL-Diff is indeed to improve the image clarity over CELL-E2 which is the major limitation for CELL-E2 and prevents it from resolving finer subcellular structures other than the nucleus and the nucleolus. We used the standard image similarity metrics to compare the generated images in the test set against the ground truth (Table 1). The improvement in these metrics should correlated with biological accuracy.
> ### 5. Sequence generation.
> For NLS sequence generation, our validation through amino acid composition analysis and clustering was exactly based on our expert knowledge of NLS. In addition, we have now further validated the generated sequences using the annotation-based localization prediction model DeepLoc 2.1[4], which indicates that 78% of the generated sequences are recognized as legitimate NLSs (Reviewer nmZP, Comment 1). Furthermore, our recent wet lab experiments testing 20 generated sequences confirmed that 10 of them exhibited NLS activity, providing direct experimental validation.
>
> [1] DeepLoc: prediction of protein subcellular localization using deep learning.
>
> [2] MultiLoc: prediction of protein subcellular localization using N-terminal targeting sequences, sequence motifs and amino acid composition.
>
> [3] Self-supervised deep learning encodes high-resolution features of protein subcellular localization.
>
> [4] DeepLoc 2.1: multi-label membrane protein type prediction using protein language models.

---

> > ### Comment · Reviewer_igYh · 2025-04-03
> >
> > Thank you for the detailed rebuttal and additional experimental validation. I appreciate the effort to clarify the intended scope of CELL-Diff and its potential applications.
> >
> > That said, I remain unconvinced on several key points. If subcellular localization is largely determined by known sequence motifs (e.g., NLS, signal peptides), the problem could arguably be framed more effectively as a classification task over discrete localization patterns — a well-established and more tractable approach. It remains unclear what scientific value is gained by generating microscopy images rather than predicting localization directly, especially given the challenges and ambiguities in mapping sequences to continuous image space.
> >
> > The claim that the HPA dataset covers over 60% of human proteins is noted, but each protein can exhibit a wide range of visual phenotypes depending on cellular context, conformational state, interactions, and post-translational modifications. This undermines the assumption of stable, predictable mappings between sequences and images.
> >
> > Crucially, I still see no evidence that the model generalizes beyond the training distribution. The authors do not explain how they prevent significant overlap between training and test proteins — a major concern, especially in light of potential memorization. While the wet-lab validation is promising, it is unclear whether these results are published or peer-reviewed, and how the tested sequences were selected. If they are close to the training distribution, this may further reinforce the concern of overfitting rather than generalization.
> >
> > Finally, while improving image clarity is a stated goal, the biological utility of generating sharper synthetic images — as opposed to interpretable or validated biological outputs — remains questionable in the absence of a clearly defined downstream application.
> >
> > For these reasons, I do not see sufficient justification to change my original rating.

---

> > > ### Author Response · Authors · 2025-04-07
> > >
> > > We respectfully disagree with the reviewer’s questioning of the premise of sequence-to-image mapping, particularly considering that this is a field that has already seen multiple peer-reviewed papers as well as preprints in the past couple of years [1,2,3,4].
> > >
> > > We appreciate the reviewer’s recognition that, unlike protein structural models for which every atomic coordinate matters, the meaningful image state space for cell biology is not at the pixel-to-pixel level. The reviewer is correct that, mechanistically, the subcellular localization of a specific protein molecule is determined by factors such as its post-translational modification, its interaction with other partners, and the overall state of the cell. A cellular image used in our training data and generated by CELL-Diff, however, describes the expected localization of all molecules for a given protein under the implicit post-translational modification and interaction profiles at the resting state of a widely used cell line, a state and a model system ubiquitously used and known to be highly generalizable in cell biology research. In this sense, the image state space is much more deterministic and thus feasibly predictable, particularly with the help of pretrained protein language embeddings. For example, RAS protein can undergo reversible lipid modification and thus dynamically cycles between the plasma membrane and cytoplasm. The result, on the other hand, is a rather defined participation coefficient between these two subcellular localizations in an image.
> > >
> > > Regarding the advantage of using image representations instead of discrete annotations for subcellular protein localization, the reviewer missed the explanations in our previous response. We would like to reiterate that image representations enable the description of multi- and variable localization patterns that are challenging for discrete annotations. This point is illustrated by the capability of CELL-Diff to characterize the ”strength” of nuclear localization and nuclear export signals using the nuclear-to-cytoplasmic ratio from segmented images instead of having to label the protein binarily as either “nuclear” or “cytoplasmic”. No existing localization annotation database has this gray-scale information. Moreover, the enhanced resolution of CELL-Diff also allows distinguishing proteins with subtle localization differences without being confined by pre-existing annotations. For example, Figure 8 shows the image prediction of two proteins, DDX5 and H1-0, both annotated to be “nucleoplasmic”. However, examining their image correlation with DNA staining reveals the specific enrichment of DDX5 at euchromatin, demonstrating the potential application of CELL-Diff in biological discovery.
> > >
> > > Finally, in response to the reviewer’s question regarding generation and sequence homology between the training and test data, we note that the yeast NLS test (see the response to Reviewer sVh2, 1. The Dataset Size and Model Generalizability) was picked to avoid any sequence homology with human proteins in the training data. Specifically, the test sequences consist of fusions between GFP (from jellyfish) and yeast sequences. Homology assessments using the Basic Local Alignment Search Tool (BLAST) confirmed that these sequences share no significant similarity with the training data (E-value $> 10^{-5}$). This setup minimizes the risk of memorization and instead requires CELL-Diff to generalize the biological principles of nuclear import recognition. The ability of the model to make accurate predictions on these out-of-distribution sequences demonstrates its capacity to generalize beyond the human proteome.
> > >
> > > [1] Khwaja, Emaad, et al. "CELL-E: A Text-to-Image Transformer for Protein Image Prediction." International Conference on Research in Computational Molecular Biology. Cham: Springer Nature Switzerland, 2024.
> > >
> > > [2] Khwaja, Emaad, et al. "CELL-E2: Translating proteins to pictures and back with a bidirectional text-to-image transformer." Advances in neural information processing systems 36 (2023): 4899-4914.
> > >
> > > [3] Zhang, Xinyi, et al. "Prediction of protein subcellular localization in single cells." bioRxiv (2024).
> > >
> > > [4] Kilgore, Henry R., et al. "Protein codes promote selective subcellular compartmentalization." Science (2025): eadq2634.

---

### Official Review · Reviewer_HAqj · 2025-03-18

**Overall Recommendation:** 5

**Summary:**

The paper, Bridging Protein Sequences and Microscopy Images with Unified Diffusion Models, presents CELL-Diff, a novel generative model that enables bidirectional transformations between protein sequences and fluorescence microscopy images. By leveraging a transformer-based U-Net architecture and integrating both continuous and discrete diffusion processes, CELL-Diff outperforms prior methods in generating high-resolution protein images. The model is trained on the Human Protein Atlas (HPA) dataset and fine-tuned on OpenCell, demonstrating its ability to reconstruct subcellular protein localization with improved fidelity. The proposed approach has significant implications for biomedical research, particularly in protein function prediction and cellular imaging.

**Claims And Evidence:**

The paper claims that CELL-Diff facilitates accurate bidirectional transformation between protein sequences and their corresponding microscopy images, improving upon previous methods like CELL-E and CELL-E2. Experimental results support this claim, demonstrating that CELL-Diff produces higher-resolution images with better spatial fidelity. The authors provide quantitative metrics such as Maximum Spatial Frequency (MSF) resolvability, Intersection over Union (IoU), and Frechet Inception Distance (FID), all of which indicate that CELL-Diff outperforms baselines.

**Essential References Not Discussed:**

The paper covers relevant prior work but does not discuss alternative generative approaches, such as GAN-based models, which have also been applied to biological image synthesis. Including a comparison with these methods could provide a broader context for CELL-Diff’s contributions.

**Experimental Designs Or Analyses:**

The experiments are well-structured, with evaluations on multiple datasets and comparisons against prior work. The use of multiple quantitative metrics strengthens the findings. However, the potential for dataset biases or domain shifts between HPA and OpenCell is not explicitly explored.

**Methods And Evaluation Criteria:**

The methodology is well-defined, employing diffusion models in both continuous (for images) and discrete (for sequences) state spaces. The evaluation framework includes comparisons with prior models (CELL-E2) using established quantitative metrics.

**Other Comments Or Suggestions:**

N/A

**Other Strengths And Weaknesses:**

Strengths:

- Introduces an innovative bidirectional generative model for protein sequences and microscopy images.
- Demonstrates significant improvements over previous methods in image quality and sequence prediction.
- Uses well-established benchmarks and evaluation metrics.

Weaknesses:

- Limited discussion on potential biases in dataset selection and domain adaptation.

**Questions For Authors:**

1. How does CELL-Diff handle proteins with highly disordered or ambiguous subcellular localization?
2. Were any domain adaptation techniques used to mitigate differences between HPA and OpenCell datasets?
3. Could alternative generative architectures, such as GANs, be competitive with the proposed approach?

**Relation To Broader Scientific Literature:**

This work aligns with research on multimodal generative modeling, fluorescence microscopy, and protein function prediction. It extends prior work on text-to-image generation by applying diffusion models to biological data. The references to related work in protein structure prediction (e.g., AlphaFold) and generative models are appropriate.

**Theoretical Claims:**

The paper does not introduce new theoretical developments but builds upon existing diffusion models. The combination of continuous and discrete diffusion processes is well-motivated.

---

> ### Author Rebuttal · Authors · 2025-03-31
>
> **Comment 1:** The potential for dataset biases or domain shifts between HPA and OpenCell is not explicitly explored.
>
> **Response:**
> The main difference between HPA and OpenCell is that HPA is larger, but OpenCell has more consistent labeling and higher image quality. The effect of domain shift has already been explored in the previous CELL-E2 paper by testing different pretraining and finetuning arrangements using the same two datasets. Therefore, we did not repeat this evaluation for CELL-Diff.
>
> ---
>
> **Comment 2:** The paper covers relevant prior work but does not discuss alternative generative approaches, such as GAN-based models, which have also been applied to biological image synthesis. Including a comparison with these methods could provide a broader context for CELL-Diff’s contributions.
>
> **Response:**
> Thank you for the comment. The baseline model used in our comparison, CELL-E2, is based on a VQGAN architecture, making it a GAN-related model. We agree that comparing CELL-Diff with alternative generative approaches would provide valuable context. However, these existing generative models are not specifically designed for this task and require significant adaptation. Implementing and optimizing these models involves careful tuning of hyperparameters and architectural modifications to achieve their best performance.
>
> We are currently exploring other generative approaches, including the consistency model[1], and have obtained some preliminary results, see the following Table. We did not observe a consistent trend of difference across metrics. We hope that the growing interest in biological image synthesis will lead to the development of more specialized models, providing a richer set of baselines for comprehensive evaluation.
>
> **Table 1: Comparison with consistency model.**
>
> | Method    | Cell image     | MSFR (nm) ↓ | IoU ↑   | FID-T ↓  | FID-O ↓  |
> |-----------|----------------|-------------|---------|----------|----------|
> | CM        | Nucl           | 650         | 0.430   | 51.3     | 38.6     |
> | CELL-Diff | Nucl           | 641         | 0.484   | 60.1     | 51.1     |
> | CM        | Nucl,ER        | 644         | 0.609   | 44.7     | 30.9     |
> | CELL-Diff | Nucl,ER        | 642         | 0.580   | 55.9     | 60.0     |
> | CM        | Nucl,MT        | 645         | 0.606   | 45.4     | 31.5     |
> | CELL-Diff | Nucl,MT        | 644         | 0.616   | 51.0     | 47.6     |
> | CM        | Nucl,ER,MT     | 645         | 0.619   | 44.6     | 31.8     |
> | CELL-Diff | Nucl,ER,MT     | 644         | 0.635   | 50.4     | 45.6     |
>
> ---
>
> **Comment 3:** Limited discussion on potential biases in dataset selection and domain adaptation.
>
> **Response:**
> See Comment 1.
>
> ---
>
> **Comment 4:** How does CELL-Diff handle proteins with highly disordered or ambiguous subcellular localization?
>
> **Response:**
> CELL-Diff handles proteins with highly disordered or ambiguous subcellular localization by leveraging the conditional cellular morphology image and the stochastic nature of the diffusion model.
>
> The cellular morphology image provides structural context, capturing the cellular environment, such as the nucleus, ER, and microtubules, as well as the cell type and cell state implicitly contained in the morphological information. This conditioning image acts as a spatial prior, guiding the model to place proteins in realistic and biologically plausible locations.
>
> Additionally, the diffusion process naturally captures stochastic variations, enabling CELL-Diff to model the inherent uncertainty of disordered or ambiguously localized proteins. As a result, the model can generate multiple plausible localizations for the same sequence when sampled multiple times, reflecting the biological variability of such proteins.
>
> ---
>
> **Comment 5:** Were any domain adaptation techniques used to mitigate differences between HPA and OpenCell datasets?
>
> **Response:**
> We currently address the domain shift between the HPA and OpenCell datasets by using the pre-training and fine-tuning approach. However, we recognize that more advanced domain adaptation techniques could further improve integration between the two datasets. Moving forward, we plan to explore additional domain adaptation methods to mitigate the differences and better align the datasets for improved model performance.
>
> ---
>
> **Comment 6:** Could alternative generative architectures, such as GANs, be competitive with the proposed approach?
>
> **Response:**
> See Comment 2.
>
> [1] Song, Yang, et al. "Consistency models." (2023).

---

> > ### Comment · Reviewer_HAqj · 2025-04-08
> >
> > I thank the authors for their answers and clarifications. I believe this work has a lot of potential practical usage : Using the cell’s visual traits to generate protein localisations is a daring but extremely biologically useful usecase. While the task is daring and very complex, the evaluation used in the paper is very sound, and contributes toward solving this hard task.
> >
> > Given the practical relevance of the task, no matter how conceptually hard it may be to achieve, and the soundness of the method and the evaluation proposed, I update my recommendation from Accept to Strong Accept.

---

### Decision · Program_Chairs · 2025-05-01

**Decision:**

Accept (poster)

**Comment:**

This paper introduces CELL-Diff, a unified diffusion model for bidirectional generation between protein sequences and fluorescence microscopy images. It combines continuous diffusion for image synthesis and discrete diffusion for sequence prediction, using a transformer-based architecture with cross-attention to integrate multimodal information. The model was trained on the Human Protein Atlas and fine-tuned on the OpenCell dataset, and it significantly improves image fidelity and protein localization. While many reviewers appreciate its potential practical values, one reviewer raised concerns on the small size of datasets. In particular, the reviewer commented that the protein localization, which depends on cell type, environment, and post-translational modifications can be very complex, and the sequence-to-image mapping is inherently one-to-many. In such a case, there is a risk of overfitting and memorizing the training data, and there is no strong evidence of good performance on the generalization of the model to new samples. The authors emphasized that CELL-Diff doesn't aim to model all sequence-to-image mappings but focuses on "virtual experiments" involving sequence variants in well-characterized cell lines. They also explained that HPA covers over 60% of human proteins and that pretrained protein embeddings help mitigate limited sample size, and GFP + yeast NLS sequences were specifically selected to avoid homology with the training data, and BLAST was used to confirm this. These explanations addressed some of the questions raised by the reviewer. Therefore, combining the positive feedbacks from other reviewers, I recommend a weak accept.